# MGF: Mixed Gaussian Flow for Diverse Trajectory Prediction

**Jiahe Chen**[1,2*]   **Jinkun Cao**[3*†]   **Dahua Lin**[4,2,5]   **Kris Kitani**[3]   **Jiangmiao Pang**[2†]

[1] Zhejiang University   [2]Shanghai AI Laboratory   [3]Carnegie Mellon University
[4]The Chinese University of Hong Kong   [5]CPII under InnoHK

∗: co-first authors   †: co-corresponding authors

## Abstract

To predict future trajectories, the normalizing flow with a standard Gaussian prior suffers from weak diversity. The ineffectiveness comes from the conflict between the fact of asymmetric and multi-modal distribution of likely outcomes and symmetric and single-modal original distribution and supervision losses. Instead, we propose constructing a mixed Gaussian prior for a normalizing flow model for trajectory prediction. The prior is constructed by analyzing the trajectory patterns in the training samples without requiring extra annotations while showing better expressiveness and being multi-modal and asymmetric. Besides diversity, it also provides better controllability for probabilistic trajectory generation. We name our method Mixed Gaussian Flow (MGF). It achieves state-of-the-art performance in the evaluation of both trajectory alignment and diversity on the popular UCY/ETH and SDD datasets. Code is available at https://github.com/mulplue/MGF.

## 1   Introduction

In this work, we aim to improve the diversity for probabilistic trajectory prediction. In trajectory prediction, diversity describes the fact that agents (pedestrians) can move in different directions, speeds, and interact with other agents. Because the motion intentions of agents can not be determined by their historical positions, there is typically no global-optimal strategy to predict a single outcome of future trajectories. Therefore, recent works have focused on probabilistic methods to generate multiple likely outcomes. However, existing solutions are argued to lack good diversity and they often fail to generate the under-represented future trajectory patterns in the training data.

Different motion patterns are usually imbalanced in a dataset. For example, agents are more likely to move straight than turn around in most datasets. Thus, many motion patterns are highly under-represented though discoverable. Therefore, intuitively, an ideal distribution to represent the possible future trajectories should be asymmetric, multi-modal, and expressive to represent long-tailed patterns.

However, most existing generative models solve the problem of trajectory prediction by modeling it as a single-modal and symmetric distribution, i.e., standard Gaussian. This is because the standard Gaussian is tractable and there is a belief that it can be transformed into any desired distribution of the same or a lower dimension. However, deriving a desired complex distribution from a simple and symmetric prior distribution is challenging, especially with limited and imbalanced data. Moreover, when we derive the target distribution by transforming from the tractable original distribution as Normalizing Flows, GANs, and VAEs do, a dilemma arises: an over-smoothing transformation model can neglect under-represented samples while an over-decorated transformation model will overfit. Especially for normalizing flow, some studies[22, 3] discussed the difficulty of training normalizing flow in practice to represent a complex target distribution.

To solve this dilemma, we propose a prior distribution with more expressiveness and data-driven statistics. It is asymmetric, multi-modal, and adaptive to the training data in the form of a mixed set of Gaussians. Compared to the standard Gaussian, the mixture of Gaussians can better summarize the under-represented samples scattered far away in the representation space. This relieves the sparsity issue of rare cases and thus enhances the diversity of the generated outcomes. Besides diversity, as

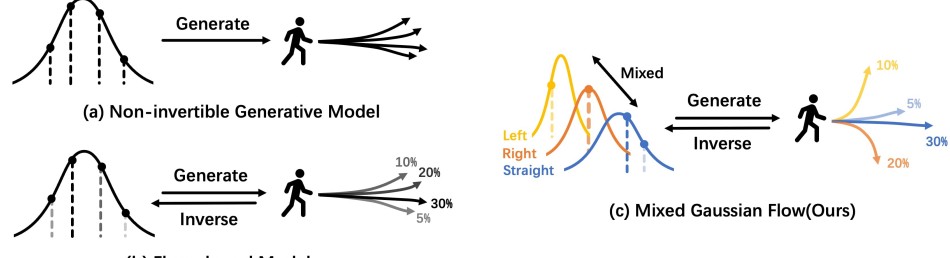

Figure 1: Non-invertible generative models (a), e.g., CVAE, GAN, and diffusions, lack the invertibility for probability density estimation. Flow-based methods (b) are invertible while, sampling from the symmetric standard Gaussian, undermines the diversity and controllability of generation. Our proposed Mixed Gaussian flow (c) maps from a mixed Gaussian prior instead. Summarizing distributions from data and controllable edits, it achieves better **diversity** and **controllability** for trajectory prediction.

the mixed Gaussian prior is parametric and transparent during construction, we could control the generation by manipulating this prior, such as adjusting the weights of different sub-Gaussians or manipulating the mean value of them. All these manipulations change generation results statistically without requiring fine-tuning or other re-training. Upon the prior distribution, we choose to construct the generative model by normalizing flow with the unique advantage of being invertible. We thus could estimate the likelihood of each single generated outcome. By combining the designs, we propose a normalizing flow-based trajectory prediction model named Mixed Gaussian Flow (MGF). It enjoys better diversity, controllability, interruptibility, and invertibility for trajectory prediction. During our study, we find that though several evaluation tools have been proposed for measuring diversity[49, 62, 33, 37], they employ varying calculation method and have not gained widespread adoption within the research community. The most popular evaluation metrics (APD/FDE scores) focus on how similar a generated trajectory is to the single ground truth. It is calculated in a "best-of-$M$" fashion where only one candidate in a batch of $M$ predictions is taken into the measurement. This protocol encourages the methods to generate outcomes approaching the mean (most likelihood) of the learned distribution and provides no sense of the diversity of generation outcomes. Therefore, building upon previous research in the field of human motion prediction[62], we formulate a metric set of Average Pairwise Displacement (APD) and Final Pairwise Displacement (FPD), which measure the diversity of a batch of $M$ generated samples. This helps us to have a concrete study about generation diversity and avoid bias from the "best-of-$M$" evaluation protocol. With the proposed metrics, we demonstrate that the proposed architecture design improves the diversity of generated trajectories. Still, we estimate the "best-of-$M$" candidate's alignment with the ground truth under widely adopted APD/FDE metrics. Surprisingly, MGF also achieves state-of-the-art performance.

To conclude, In this work, we focus on enhancing the diversity of trajectory prediction. We propose Mixed Gaussian Flow (MGF) by reforming the prior distribution of normalizing flows as a novel design of mixed Gaussians. It achieves state-of-the-art performance with respect to both the "best-of-$M$" alignment metrics and diversity metrics. We demonstrate that the proposed MGF model is capable of diverse and controllable trajectory predictions.

## 2 Related Works

**Generative Models for Trajectory Prediction.** Trajectory prediction aims to predict the positions in a future horizon given historical position observations of multiple participants (agents). Early studies solve the problem by deterministic trajectory prediction [20] where Social forces [14], RNNs [1, 35, 52], and the Gaussian Process [53] are proposed to model the agent motion intentions. Recent works mostly seek multi-modal and probabilistic solutions for trajectory prediction instead, which is a more challenging but faithful setting. Though some of them leverage reinforcement learning [25, 4], the mainstream uses generative models to solve the problem as generating likely future trajectories. Auto-encoder [18] and its variants, such as CVAE [23, 63], are widely adopted. GANs make another line of work [13]. More recently, the diffusion [15] model is also used in this area [31]. However, they are typically not capable of estimating outcome probability as the generation process is not invertible. Normalizing flow [21] is preferred in many cases for being invertible.

**Normalizing Flow for Trajectory Prediction.** In this work, we would like the predicted trajectories diverse and controllable. We prefer the generation process invertible to allow tractable generation likelihood. We thus choose normalizing flow [21] generative models. Normalizing flow [36] constructs complex distributions by transforming a probability density through invertible mappings

from tractable distribution. Normalizing flow has been studied for trajectory prediction in some previous works [40, 41, 12]. In the commonly adopted evaluation protocol of "best-of-$M$" trajectory candidates, normalizing flow-based methods are once considered not capable of achieving state-of-the-art performance. However, we will show in this paper that with proper design of architecture, normalizing flow can be state-of-the-art. And much more importantly, its invertibility allows more controllable and explainable trajectory prediction.

**Gaussian Mixture models as prior.** Though the standard Gaussian is chosen by mainstream generative models as the original sampling distribution, some previous works explored how Gaussian mixture models (GMM) can be an alternative to help with generation or classification tasks. [8] uses a GMM prior in VAE models to enhance the clustering and classification performance. [2] adopts GMM to enhance the conditional generation of GAN networks. FlowGMM [17] uses GMM as the prior for flow-based models to deal with the classification task in a semi-supervised way. A recent work PluGen [54] proposes to mix two Gaussians to model the conditional presence of two binary labels to control generation tasks. Existing methods mostly use GMM to describe the presence of multiple auxiliary labels and they typically require additional annotations to construct the GMM. In this work, we use GMM as the distribution prior for normalizing flows without requiring any label annotations. It is designed to enhance the diversity of the generation and relieve the difficulty of learning transforming the tractable prior distribution to the desired complex and multi-modal target distribution for future trajectory generation.

## 3 Method

Our proposed method is based on the normalizing flow paradigm for invertible trajectory generation while we construct a mixed Gaussian prior as the original distribution instead of the naive standard Gaussian to allow more diverse and controllable outcomes. In this section, we first provide the formal problem formulation in Section 3.1. Then we introduce normalizing flow in Section 3.2 and the proposed Mixed Gaussian Flow (MGF) model in Section 3.3. We detail the training/inference implementations in Section 3.4. At last, we introduce the proposed metrics set to measure the diversity of generated trajectories in Section 3.5. The overall illustration of MGF is shown in Figure 2.

### 3.1 Problem Formulation

We focus on 2D agent trajectory forecasting and represent the agent positions by 2D coordinates. Given a set of multiple agents, i.e., pedestrians in our case, we denote the 2D position of an agent $a$ at time $t$ as $\mathbf{x}_t^a$ and a trajectory from $t_i$ to $t_j(t_i < t_j)$ as $\mathbf{x}_{t_i:t_j}^a$. Given a fixed scene with map $\mathbf{M}$ and a period $\mathbf{T}$: $t_0, t_1, t_2, ..., t_c, ..., t_T$, there are $N$ agents that have appeared during the period $\mathbf{T}$, denoted as $A_{t_0:t_T} = \{a_0, a_1, ..., a_{N-1}\}$. Without loss of generality, given a current time step $t_c \in (t_0, t_T)$, the task of trajectory prediction aims to obtain a set of likely trajectories $\mathbf{x}_{t_c:t_T}^a$ with the past trajectories of all observed agents $\mathbf{X}_{t_0:t_c}^{A_{t_0:t_c}} = \{\mathbf{x}_{t_0:t_c}^a, a \in A_{t_0:t_c}\}$ as input, where $a$ is an arbitrary agent that has shown up during $t : t_0 \longrightarrow t_c$. For each agent $a \in A_{t_0:t_c}$ we seek to sample plausible and likely trajectories of it over the remaining time steps $t_c \longrightarrow t_T$ by a generative model $\Phi$, i.e.,

$$\hat{\mathbf{x}}_{t_c:t_T}^a = \Phi(\mathbf{X}_{t_0:t_c}^{A_{t_0:t_c}}), \tag{1}$$

at the same time, when there are other variables such as the observations of the maps are provided, we can use them as additional input information. By denoting the observations until $t$ as $\mathbf{O}_{t_0:t_c}$ we have

$$\hat{\mathbf{x}}_{t_c:t_T}^a = \Phi(\mathbf{x}_{t_0:t_c}^{A_{t_0:t_c}}; \mathbf{O}_{t_0:t_c}). \tag{2}$$

If the generation process is probabilistic instead of deterministic, the outcome of the solution is a set of trajectories instead of a single one. The formulation thus turns to

$$\{^{(i)}\hat{\mathbf{x}}_{t_c:t_T}^a\} = \Phi(\mathbf{x}_{t_0:t_c}^{A_{t_0:t_c}}; \mathbf{O}_{t_0:t_c}), \tag{3}$$

where $i$ is the index of one candidate in the predicted batch.

For some generative models relying on transforming from a sample point in a known distribution $\mathcal{D}_0$ to the target distribution, e.g., GANs and normalizing flows, the set is generated by mapping from different sample points, i.e., $p \in \mathcal{D}_0$. Therefore, the full formulation becomes

$$\{^{(i)}\hat{\mathbf{x}}_{t_c:t_T}^a\} = \Phi(\mathbf{x}_{t_0:t_c}^{A_{t_0:t_c}}; \mathbf{O}_{t_0:t_c}, \mathbf{P}), \tag{4}$$

where $\mathbf{P} = \{p_0, ..., p_K\}$ is a set of sampled points from $\mathcal{D}_0$.

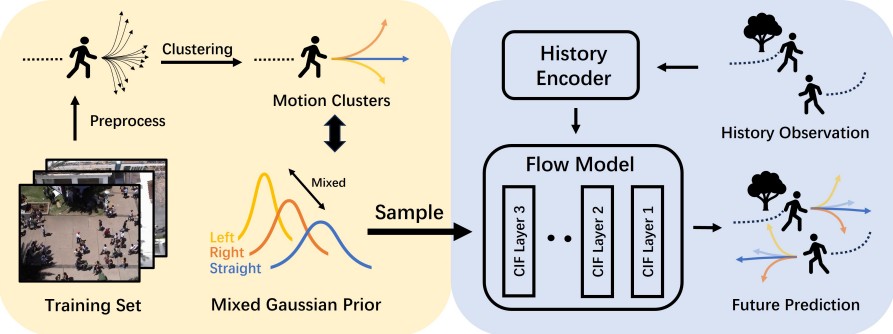

| Mixed Gaussian Prior Construction | Flow Prediction |

Figure 2: The illustration of our proposed Mixed Gaussian Flow (MGF). During training, we construct a mixed Gaussian prior by statistics from the training set. During sampling, the initial noise samples are from the constructed mixed Gaussian prior. MGF keeps a tractable prior distribution and an invertible inference process while the novel mixed Gaussian prior provides more diversity and controllability to the generation outcomes.

Implicitly, the model $\Phi$ is required to construct a transformation (Jacobians) between the two distributions. Usually, $\mathcal{D}_0$ is chosen as a symmetric and tractable distribution, such as a standard Gaussian. However, the distribution of the target distribution can be shaped by many data-biased asymmetries thus posing a challenge to learning the transformation effectively and inclusively. This often causes failure of generating under-represented trajectory patterns for trajectory forecasting and hurts the diversity of the outcomes. This observation motivates us to propose a probabilistic generative model for more diverse outcomes by representing the original distribution with more expressiveness.

### 3.2 Normaling Flow

Normalizing flow [21] is a genre of generative model that constructs complex distributions by transforming a simple distribution through a series of invertible mappings. We choose normalizing flow over other probabilistic generative models as it can provide per-sample likelihood estimates thanks to being invertible. This property is critical to more comprehensively understand the distribution of different future trajectory patterns, especially when typically only sampling dozens of outcomes and considering the existence of long-tailed trajectory patterns. We denote a normalizing flow as a bijective mapping $f$ which transforms a simple distribution $p(\mathbf{z})$ to a complex distribution $p(\mathbf{x})$. The transformation is often conditioned on context information $\mathbf{c}$. With the change-of-variables formula, we can derive the transformation connecting two smooth distributions as follows:

$$
\begin{aligned}
\mathbf{x} &= f(\mathbf{z}; \mathbf{c}), \\
p(\mathbf{x}) &= p(\mathbf{z}) \cdot |\det(\nabla_{\mathbf{x}} f^{-1}(\mathbf{x}; \mathbf{c}))|, \\
-\log(p(\mathbf{x})) &= -\log(p(\mathbf{z})) - \log(|\det(\nabla_{\mathbf{x}} f^{-1}(\mathbf{x}; \mathbf{c}))|).
\end{aligned}
\tag{5}
$$

Given the formulations, with a known distribution $\mathbf{z} \sim \mathcal{D}_0$, we can calculate the density of $p(\mathbf{x})$ following the transformations and vice versa. However, the equations require the Jacobian determinant of the function $f$ to obtain the distribution density $p(\mathbf{x})$. The calculation of it in the high-dimensional space is not trivial. Recent works propose to use deep neural networks to approximate the Jacobians. To maintain the inevitability of the normalizing flows, some carefully designed layers are inserted into the deep models and the coupling layers [9] are one of the most widely adopted ones.

More recently, FlowChain [28] is proposed to enhance the standard normalizing flow models by using a series of Conditional Continuously-indexed Flows (CIFs) [5] to estimate the density of outcomes. CIFs are obtained by replacing the single bijection $f$ in normalizing flows with an indexed family $F(\cdot; u)_{u \in U}$, where $U \subseteq R$ is the index set and each $F(\cdot; u) : \mathbf{z} \longrightarrow \mathbf{x}$ is a bijection. Then, the transformation is changed to

$$
\mathbf{z} \sim p(\mathbf{z}), \quad U \sim p_{U|\mathbf{z}}(\cdot|\mathbf{z}), \quad \mathbf{x} := F(\mathbf{z}; U). \tag{6}
$$

Please refer to [5] for more details about CIFs and their connection with variational inference. In this work, we follow the idea of using a stack of CIFs from [28] to achieve fast inference and the updates of trajectory density estimates.

Normalizing flow based model samples from a standard Gaussian, $\mathbf{z} \sim \mathcal{N}(0, 1)$, usually results in overfitting to the most-likelihood for trajectory prediction. This is because each data sample from the

training sample is considered extracted as the mode of a standard Gaussian. Only the mode value (the ground truth) is directly supervised and the underlying target distribution is assumed to be perfectly symmetric, which is not aligned with the usual real-world facts. Related discussion can be found in many previous literatures[43, 19]. This typically results in degraded expressiveness of the model to fail to capture under-represented motion patterns from the data and thus hurts the outcome diversity.

### 3.3 Mixed Gaussian Flow (MGF)

We propose Mixed Gaussian Flow (MGF) to enhance the diversity and controllability in trajectory prediction. MGF consists of two stages as summarized in Figure 2. First, we construct the mixed Gaussian prior by fitting the parametric model of a combination of $K$ Gaussians, $\{\mathcal{N}(\mu_k, \sigma_k^2)\}, (1 \leq k \leq K)$. The parametric model is obtained with the data samples from training sets. Then, during inference, we sample points from the mixture of Gaussian and map them into a trajectory latent in the target distribution by a stack of CIF layers with the historical trajectories of all involved agents as the condition. We will introduce the two stages in detail below.

MGF maps from a mixture of Gaussians instead of a single Gaussian to the target distribution. To maintain the inevitability of the model, the mixed Gaussian prior can not be arbitrary. We obtain the parametric construction of the mixed Gaussian by fitting it with training data. In this fashion, we can derive multiple Gaussians to represent different motion patterns in the dataset, such as going straight or turning left and right. In a simplified perspective, we regard the mixture as combining multiple clusters, each of which represents a certain sub-distribution. By sampling from the mixture of Gaussians instead of a standard Gaussian, our constructed model has more powerful expressiveness than the standard normalizing flow model. This results in more diverse trajectory predictions. Also, by manipulating the mixed Gaussian prior, we can achieve controllable trajectory prediction.

**Mixed Gaussian Prior Construction.** For the data pre-processing, we transfer motion directions into relative directions with respect to a zero-degree direction. All position footage is represented in meters. Given the trajectory between $t_0 \longrightarrow t_c$ to predict the trajectory between $t_c \longrightarrow t_T$, we would put the position pivot at $t_c$, i.e., $\mathbf{x}_{t_c}$, as the origin and convert the position on all other time steps to be the offset from $\mathbf{x}_{t_c}$. Then, we cluster the preprocessed future trajectories into $K$ clusters, which is a hyper-parameter. We note the mean of the clusters as $\boldsymbol{\mu} = \{\mu_i\}_{i=1,...,K}$.

These cluster centers reveal the mean value of $K$ representative patterns of pedestrians' motion, e.g. go straight, turn left. They will be the means of the Gaussians. The variances of the Gaussian, i.e., $\sigma_k^2$, can be pre-determined or learned. The final mixture of Gaussians is denoted as

$$\mathcal{D}^\Sigma = \sum_{k=1}^{K} \beta_k \mathcal{N}(\mu_k, \sigma_k^2), \tag{7}$$

where $\beta_k$ are the weights assigned to each cluster following the k-means clustering of the training data. By default, we perform clustering by K-means with $K = 8$.

**Flow Prediction.** Once the mixed Gaussian prior is built, we can do trajectory prediction by mapping samples from the distribution to future trajectories conditioned on historical information(e.g. social interaction features extracted by a Trajectron++[46] encoder). Here, we ignore the intermediate transformation by CIFs as Equation (6) shows while following the original formulations of normalizing flows as Equation (5) for simplicity. We distribute the samples from different Gaussians by their weights. Given the $i$-th sample from $\mathcal{N}(\mu_k, \sigma_k^2)$, we can transform it to the $i$-th predicted trajectories

$$\mathbf{z}_i \sim \mathcal{D}^\Sigma, \quad {}^{(i)}\hat{\mathbf{x}}_{t_c:t_T}^a = \Phi(\mathbf{x}_{t_0:t_c}^{A_{t_0:t_c}}; \mathbf{O}_{t_0:t_c}, \mathbf{z}_i). \tag{8}$$

For a sample $\frac{\mathbf{z}_i}{\beta_k} \sim \mathcal{N}(\mu_k, \sigma_k^2)$, we have the probability estimate

$$p(\mathbf{z}_i) = \beta_k \frac{1}{\sigma_k \sqrt{2\pi}} e^{-\frac{(z_i - \mu_k)^2}{2\sigma_k^2}}, \tag{9}$$

and the transformation is converted to

$$p({}^{(i)}\hat{\mathbf{x}}_{t_c:t_T}^a) = \exp(-\frac{(z_i - \mu_k)^2}{2\sigma_k^2} + \log \frac{\beta_k}{\sigma_k \sqrt{2\pi}}) \cdot |\det(\nabla_{f(\mathbf{z_i};\mathbf{O}_{t_0:t_c})} \mathbf{z_i})|, \tag{10}$$

which can be also invested back for the density estimate by the normalizing flow law

$$\hat{p}(\mathbf{z}_i) = \beta_k \frac{1}{\sigma_k \sqrt{2\pi}} \exp(-\frac{[f^{-1}({}^{(i)}\hat{x}_{t_c:t_T}^a; O_{t_0:t_c}) - \mu_k]^2}{2\sigma_k^2}). \tag{11}$$

### 3.4 Training and Inference

The training loss of MGF comes from two directions: the forward process to get mixed flow loss and the inverse process to get minimum $\ell_2$ loss.

**Forward process.** Given a ground truth trajectory sample $\mathbf{x}^a_{t_c:t_T}$, we need to assign it to a cluster in the mixed Gaussian prior by measuring its distance to the centroids

$$\hat{k} = \arg\min_i (\mathbf{x}^a_{t_c:t_T} - \mu_i)^2, \quad \mathcal{D}^{\hat{k}} := \beta_{\hat{k}} \mathcal{N}(\mu_{\hat{k}}, \sigma_{\hat{k}^2}), \tag{12}$$

with a tractable probability density function $p_{\hat{k}}(\cdot)$. Through the inverse process $f^{-1}$ of flow model, we transform $\mathbf{x}^a_{t_c:t_T}$ into its corresponding latent representation, here denoted as

$$\hat{\mathbf{z}} = f^{-1}(\mathbf{x}^a_{t_c:t_T}; \mathbf{O}_{t_0:t_c}). \tag{13}$$

Then we can compute the forward mixed flow loss:

$$L_{forward} = -\log(p(\mathbf{x}^a_{t_c:t_T})) = -\log(p_{\hat{k}}(\hat{\mathbf{z}})) - \log(|\det(\nabla_{\mathbf{x}^a_{t_c:t_T}} \hat{\mathbf{z}})|). \tag{14}$$

Instead of computing negative-log-likelihood(NLL) loss of $\hat{\mathbf{z}}$ in the mixed distribution $\sum_{k=1}^{K} \beta_k \mathcal{N}(\mu_k, \sigma_k^2)$, we compute NLL loss in the sub-Gaussian with the nearest centroid $\beta_{\hat{k}} \mathcal{N}(\mu_{\hat{k}}, \sigma_{\hat{k}^2})$ because each centroid is independent to others in the mixed distribution and we encourage the model to learn specified motion patterns to avoid overwhelming by the major data patterns. Calculating NLL loss over the mixed distribution may fuse other centroids and damage the diversity of model outputs. By our design, the mixed Gaussian prior can maintain more capacity for expressing complicated multi-modal distribution than the traditional single Gaussian prior, which typically constrains the target distribution to be single-modal and symmetric.

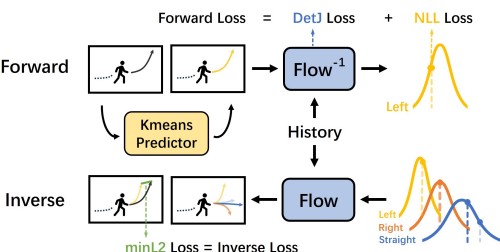

Figure 3: During training, the model is trained at both forward and inverse process of the normalizing flow.

**Inverse Process.** This process repeats the flow prediction process to get generated trajectories. To predict $M$ candidates, we sample $\mathbf{z}_i \sim \sum_{k=1}^{K} \beta_k \mathcal{N}(\mu_k, \sigma_k^2), i = 1, 2, ..., M$ and transform them into $M$ trajectories

$$\{^{(i)}\hat{\mathbf{x}}^a_{t_c:t_T}\} = \{f(\mathbf{z}_i; \mathbf{O}_{t_0:t_c})\}, i = 1, 2, ..., M. \tag{15}$$

We compute the minimum $\ell_2$ loss between $M$ predictions and ground truth trajectory as [13] does:

$$L_{inverse} = \min_{i=1}^{M} \frac{(^{(i)}\hat{\mathbf{x}}^a_{t_c:t_T} - \mathbf{x}^a_{t_c:t_T})^2}{t_T - t_c}. \tag{16}$$

We sample $\mathbf{z}_i$ from sub Gaussians by their weight. This is approximately equal to sampling from the original mixed Gaussians but makes the reparameterization trick doable.

Although approximated differential backpropagation techniques, such as the Gumbel-Softmax trick, can be employed to make the sampling process of mixed Gaussians differentiable, computing the Negative Log-Likelihood (NLL) loss between a sample point and the mixed Gaussian distribution remains challenging because

$$-\log(p_{\mathcal{D}^{\Sigma}}(\hat{\mathbf{z}})) = -\log\left(\sum_{k=1}^{K} \frac{\beta_k}{\sigma_k} \cdot e^{-\frac{(\hat{z}-\mu_k)^2}{2\sigma_k^2}}\right) + C, \tag{17}$$

contains exponential operations on matrices, which can be simplified through logarithmic operations in single Gaussian condition. Computing this term requires iterative optimization methods, such as the Expectation-Maximization algorithm[6] for approximation[60, 47], which makes the computing process much more complex. Therefore, in practice, sampling from individual Gaussian components is preferred for computing efficiency. Furthermore, applying the Gumble-softmax to learn a mixture of Gaussians in generative models has been reported difficult in practice in some cases[39] due to gradient vanishing problem.

The forward and inverse losses encourage the model to predict a well-aligned sample in a sub-space from the prior without hurting the flexibility and expressiveness of other sub-spaces. We combine the forward and inverse losses by a ratio $\gamma$ to be a Symmetric Cross-Entropy loss [40], which was proved beneficial for better balancing the "diversity" and "precision" of predicted trajectories:

$$L = L_{forward} + \gamma \cdot L_{inverse}. \tag{18}$$

### 3.5  Diversity Metrics

The widely adopted average/final displacement error (ADE/FDE) scores measure the alignment (precision) between the ground truth future trajectory and one predicted trajectory. Under the common "best-of-$M$" evaluation protocol, ADE/FDE scores encourage nothing but finding a single "aligned" trajectory with the ground truth. ADE encourages the position on all time steps to be aligned with the single ground truth and FDE chooses the trajectory with the closest endpoint while all other trajectories are neglected in score calculating. Such an evaluation protocol overwhelmingly encourages the methods to fit the most likelihood from a certain distribution and all generated candidates race to be the most similar one as the distribution mean. Under the single-mode and symmetric assumption, this usually tends to fit into a Gaussian with a smaller variance. However, this tendency hurts the diversity of predicted trajectory hypotheses.

To provide a tool for quantitative trajectory diversity evaluation, we formulate a set of metrics. Following the idea of average displacement error (ADE) and final displacement error (FDE), we measure the diversity of trajectories by their pairwise displacement along the whole generated trajectories and the final step. Follow Dlow[62], we name that average pairwise displacement (APD) and final pairwise displacement (FPD). We note that the diversity metrics are measured in the complete set of generated trajectory candidates instead of between a single candidate and the ground truth. The formulation of APD and FPD are as below

$$\text{APD} = \frac{\sum_{i=1}^{M} \sum_{j=1}^{M} \sqrt{\sum_{t=t_c}^{t_T} (^{(i)}\hat{\mathbf{x}}_t^a - ^{(j)}\hat{\mathbf{x}}_t^a)^2}}{M^2 \cdot (t_T - t_c)}, \quad \text{FPD} = \frac{\sum_{i=1}^{M} \sum_{j=1}^{M} \sqrt{(^{(i)}\hat{\mathbf{x}}_{t_T}^a - ^{(j)}\hat{\mathbf{x}}_{t_T}^a)^2}}{M^2}, \tag{19}$$

where APD measures the average displacement along the whole predicted trajectories and FPD measures the displacement of trajectory endpoints. We would mainly follow the widely adopted ADE/FDE for benchmarking purposes while using APD/FPD as a secondary metric set to better understand the diversity of the generated future trajectories.

## 4  Experiments

In this section, we provide experiments to demonstrate the effectiveness of our method. We first introduce experiment setup in Section 4.1 and benchmark with related works to evaluate the trajectory prediction alignment and diversity in Section 4.2. Then, we showcase the diversity and controllability of MGF in Section 4.3 and Section 4.4. Finally, we ablate key implementation components in Section 4.5.

### 4.1  Setup

**Datasets.** We evaluate on two major benchmarks, i.e., ETH/UCY [24, 38] and SDD [42]. ETH/UCY consists of five subsets. We follow the widely used Social-GAN [13] benchmark. SDD dataset consists of 20 scenes captured in bird's eye view. We follow the TrajNet [44] benchmark. We note that in the community of trajectory prediction, previous works have inconsistent evaluation protocol details and thus have made unfair comparisons. Please refer to the appendix in supplementary materials for details.

**Metrics.** We use the widely used average displacement error (ADE) and final displacement error (FDE) to measure the alignment of the predicted trajectories and the ground truth. ADE is the average L2 distance between the ground truth and the predicted trajectory. FDE is the L2 distance between the ground truth endpoints and predictions. Most previous works choose the "Best-of-$M$" evaluation protocol and we follow it to choose $M = 20$ as default.

Here, we note that, under different assumptions of distribution spreading and variance, the evaluation is ideally done with different values of $M$. However, most existing methods only provide results with $M = 20$ and many of them do not open-source the code of the models so we can not rebenchmark with other value choices of $M$. Besides the metrics for trajectory alignment, we also use the proposed metrics set APD and FPD to measure the diversity of the predicted trajectory candidates.

Table 1: **Results on *ETH/UCY* dataset with Best-of-20 metrics.** Scores are in meters, lower is better. **bold** and underlined scores denote the best and the second-best scores.

| Method | ETH | | HOTEL | | UNIV | | ZARA1 | | ZARA2 | | Mean | |
|---|---|---|---|---|---|---|---|---|---|---|---|---|
| | ADE | FDE | ADE | FDE | ADE | FDE | ADE | FDE | ADE | FDE | ADE | FDE |
| Social-GAN [13] | 0.87 | 1.62 | 0.67 | 1.37 | 0.76 | 1.52 | 0.35 | 0.68 | 0.42 | 0.84 | 0.61 | 1.21 |
| STGAT [16] | 0.65 | 1.12 | 0.35 | 0.66 | 0.52 | 1.10 | 0.34 | 0.69 | 0.29 | 0.60 | 0.43 | 0.83 |
| Social-STGCNN [32] | 0.64 | 1.11 | 0.49 | 0.85 | 0.44 | 0.79 | 0.34 | 0.53 | 0.30 | 0.48 | 0.44 | 0.75 |
| Trajectron++ [46] | 0.61 | 1.03 | 0.20 | 0.28 | 0.30 | 0.55 | 0.24 | 0.41 | 0.18 | 0.32 | 0.31 | 0.52 |
| MID [11] | 0.55 | 0.88 | 0.20 | 0.35 | 0.30 | 0.55 | 0.29 | 0.51 | 0.20 | 0.38 | 0.31 | 0.53 |
| PECNet [30] | 0.54 | 0.87 | 0.18 | 0.24 | 0.35 | 0.60 | 0.22 | 0.39 | 0.17 | 0.30 | 0.29 | 0.48 |
| GroupNet [57] | 0.46 | 0.73 | 0.15 | 0.25 | 0.26 | 0.49 | 0.21 | 0.39 | 0.17 | 0.33 | 0.25 | 0.44 |
| AgentFormer [63] | 0.45 | 0.75 | 0.14 | 0.22 | 0.25 | 0.45 | 0.18 | 0.30 | 0.14 | 0.24 | 0.23 | 0.39 |
| EqMotion [59] | 0.40 | 0.61 | **0.12** | **0.18** | 0.23 | 0.43 | 0.18 | 0.32 | **0.13** | **0.23** | 0.21 | 0.35 |
| FlowChain [28] | 0.55 | 0.99 | 0.20 | 0.35 | 0.29 | 0.54 | 0.22 | 0.40 | 0.20 | 0.34 | 0.29 | 0.52 |
| MGF(Ours) | **0.39** | **0.59** | 0.13 | 0.20 | **0.21** | **0.39** | **0.17** | **0.29** | 0.14 | 0.24 | **0.21** | **0.34** |

Table 3: **Results on *ETH/UCY* dataset with diversity metrics.** Scores are in meters, higher means more diverse prediction. **bold** and underlined scores denote the best and the second-best scores.

| Method | ETH | | HOTEL | | UNIV | | ZARA1 | | ZARA2 | | Mean | |
|---|---|---|---|---|---|---|---|---|---|---|---|---|
| | APD | FPD | APD | FPD | APD | FPD | APD | FPD | APD | FPD | APD | FPD |
| Social-GAN [13] | 0.680 | 1.331 | 0.566 | 1.259 | 0.657 | 1.502 | 0.617 | 1.360 | 0.515 | 1.119 | 0.607 | 1.314 |
| Social-STGCNN [32] | 0.404 | 0.633 | 0.591 | 0.923 | 0.333 | 0.497 | 0.490 | 0.762 | 0.417 | 0.657 | 0.447 | 0.694 |
| Trajectron++ [46] | 0.704 | 1.532 | 0.568 | 1.240 | 0.648 | 1.404 | 0.697 | 1.528 | 0.532 | 1.161 | 0.630 | 1.373 |
| AgentFormer [63] | **1.998** | **4.560** | 0.995 | 2.333 | 1.049 | **2.445** | 0.774 | 1.772 | 0.849 | 1.982 | 1.133 | **2.618** |
| MemoNet [58] | 1.232 | 2.870 | 0.950 | 2.030 | 0.847 | 1.822 | 0.844 | 1.919 | 0.880 | 2.120 | 0.951 | 2.152 |
| FlowChain [28] | 0.814 | 1.481 | 0.484 | 0.833 | 0.636 | 1.094 | 0.505 | 0.890 | 0.492 | 0.859 | 0.586 | 1.031 |
| MGF(Ours) | 1.624 | 3.555 | **1.138** | **2.387** | **1.115** | 2.163 | **1.029** | **2.119** | **1.065** | **2.182** | **1.194** | 2.481 |

**Implementation Details.** We enhance our model using a similar technique as "intension clustering" [58] and we name it "prediction clustering". The key difference is that we directly cluster the entire trajectory instead of the endpoints. To make a fair comparison, we followed the data processing from FlowChain [28] and Trajectron++ [46]. We also follow FlowChain's implementations of CIFs that each layer consists of a RealNVP [10] with a 3-layer MLP and 128 hidden units. We use a Trajectron++ [46] encoder to encode historical trajectories. All models were trained on a single NVIDIA V100 GPU for 100 epochs(approximately 4 to 8 hours).

Table 2: **Evaluation results on *SDD* (in pixels).**

| Method | ADE | FDE |
|---|---|---|
| Social-GAN [13] | 27.25 | 41.44 |
| STGAT [16] | 14.85 | 28.17 |
| Social-STGCNN [32] | 20.76 | 33.18 |
| Trajectron++ [46] | 19.30 | 32.70 |
| MID [11] | 10.31 | 17.37 |
| PECNet [30] | 9.97 | 15.89 |
| GroupNet [57] | 9.31 | 16.11 |
| EqMotion [59] | 8.80 | 14.35 |
| MemoNet [58] | 8.56 | 12.66 |
| FlowChain [28] | 9.93 | 17.17 |
| MGF (Ours) | **7.74** | **12.07** |

## 4.2 Benchmark Results

We benchmark MGF with a line of recent related works on ETH/UCY dataset in Table 1. The results of Trajectron++ and MID are updated according to a reported implementation issue [1]. MGF achieves on-par state-of-the-art performance with Eqmotion [59]. Specifically, Our method achieves the best ADE and FDE in 3 out of 5 subsets and the best ADE and FDE score by averaging all splits. Here we note that we build MGF as a normalizing flow-based method as its invertibility is key property we desire, though normalizing flow is usually considered inferior regarding the alignment evaluation. Therefore, such a good performance on the alignment is surprising to us. To compare with other normalizing flow-based methods, our method significantly improves the performance compared to FlowChain, achieving **27.6%** improvement by ADE and **34.6%** improvement by FDE.

On the SDD dataset, where the motion pattern is considered more diverse than UCY/ETH, the benchmark results are shown in Table 2. Our method outperforms all baselines measured by ADE/FDE for trajectory alignment. Specifically, Our method reduces ADE from 8.56 to 7.74 compared to the current state-of-the-art method MemoNet, achieving **9.6%** improvement. Our method also significantly improves the performance of FlowChain for **22.1%** by ADE and **29.7%** by FDE. According to the benchmarking on the two popular datasets, we demonstrate the state-of-the-art alignment (precision) of our proposed method. Here we note again that the alignment with the deterministic ground truth is not the highest priority when we design our method, we will discuss the main advantages of MGF, diversity, and controllability, in the next paragraphs.

---

[1]https://github.com/StanfordASL/Trajectron-plus-plus/issues/53

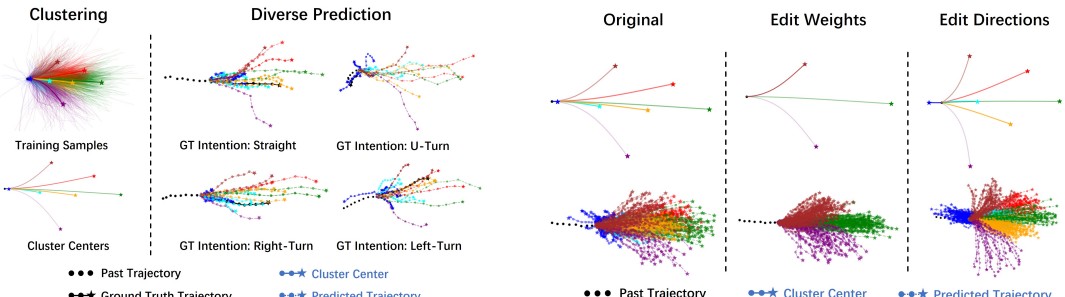

Figure 4: MGF predictions on ETH dataset. The color of trajectories corresponds to the cluster in the mixed Gaussian prior, from which the sample belongs to.

Figure 5: Controllable generation on ETH dataset. By editing cluster centers, we can control the predictions.

## 4.3 Diverse Generation

By leveraging the mixed Gaussian prior, our model can generate trajectories from the corresponding clusters, resulting in a more diverse set of trajectories than sampling from a Gaussian. This is intuitively due to less difficulty in learning the Jacobians for distribution transformation. We present examples in Figure 4. Given a past trajectory, there is a single ground truth future trajectory possibility from the dataset. We select four samples with different ground truth intentions, i.e., going straight, U-turn, left-turn, and right-turn. By sampling noise from the clustered distributions, we could generate future trajectories with diverse intentions. From the visualizations, we could notice, of course, that we generate outcomes that are very similar to the ground truth with close intentions while we also generate outcomes that have very diverse intentions. The well-aligned single trajectory accounts for the high ADE and FDE score our method achieves. And the impressive diversity demonstrates the effectiveness of our design, especially considering they are well controlled by the clusters where they are sampled from.

Quantitatively, we evaluate the generation diversity according to our proposed metrics on ETH/UCY dataset since most existing methods did not either make experiments on SDD or open-source training code/checkpoint on SDD. The results are presented in Table 3. We can observe that MGF achieves the best or second-best APD and FPD score on all splits among sota methods. Besides, our method significantly improves the performance compared to FlowChain, achieving **103.7%** improvement by APD and **140.6%** improvement by FPD. The only method that can achieve close diversity with our method is Agentformer [63], which designs sampling from a set of conditional VAE to improve the diversity. However, compared to MGF, Agentformer is more computation-intensive and shows significantly lower alignment according to ADE/FDE scores in Table 1. Also, Agentformer is not fully invertible, which is considered a key property we desire for trajectory forecasting. The superior quantitative performance according to the alignment (precision) and diversity metrics suggests the effectiveness of our method by balancing these two adversarial features.

We also find that APD/FPD metrics are not sensitive to $M$, which is a natural result, see appendix B.

## 4.4 Controllable Generation

The generated sample from MGF is highly correlated with the original sample drawn from the mixed Gaussian prior. If the prior distribution is a standard Gaussian as in the canonical normalizing flow method, we can have almost no control over the generated sample. The only controllability is to sample near the mode to generate a sample similar to the learned most-likelihood outcome or far from the mode to make them more different. However, as we discussed, after sufficient training and supervision by the forward loss, the variance of the latent Gaussian distribution of the outcome is usually very small, which further hurts the controllability. However, as we chose a transparent mixed Gaussian prior for the sampling, we can control the generation flexibility. First, by adjusting sub-Gaussians in the mixture prior, we can manipulate the generation process statistically. Figure 5 shows that by editing cluster compositions, we can control the predictions of MGF with good interpretability. By editing the weights of sub-Gaussians, we can control the ratio of splatting into directions. By editing the directions of the cluster means, we can control the intentions of samples statistically. Besides cluster centers, we can also edit the variance of Gaussian to control the density of generated trajectories or combine a set of operations to get expected predictions. We provide more discussions and examples in the appendix in the supplement.

Table 4: Ablation study of ADE/FDE on the ETH/UCY and SDD dataset.

| Pred. Clustering | Mixed Gaus. | Learnable Var. | Inv. Loss | ETH/UCY | | SDD | |
|---|---|---|---|---|---|---|---|
| | | | | **ADE** | **FDE** | **ADE** | **FDE** |
| - | - | - | - | 0.33 | 0.61 | 11.90 | 21.33 |
| - | ✓ | - | - | 0.29(↓0.04) | 0.48(↓0.13) | 11.38(↓0.52) | 19.28(↓2.05) |
| ✓ | - | - | - | 0.29 | 0.54 | 10.63 | 18.80 |
| ✓ | ✓ | - | - | 0.27(↓0.02) | 0.48(↓0.06) | 9.19(↓1.44) | 15.86(↓2.94) |
| ✓ | ✓ | ✓ | - | 0.23(↓0.04) | 0.39(↓0.09) | 8.71(↓0.48) | 14.86(↓1.00) |
| ✓ | ✓ | ✓ | ✓ | 0.21(↓0.02) | 0.34(↓0.05) | 7.74(↓0.97) | 12.07(↓2.79) |

Table 5: Ablation study of APD/FPD on the ETH/UCY and SDD dataset.

| Pred. Clustering | Mixed Gaus. | Learnable Var. | Inv. Loss | ETH/UCY | | SDD | |
|---|---|---|---|---|---|---|---|
| | | | | **APD** | **FPD** | **APD** | **FPD** |
| - | - | - | - | 0.39 | 0.76 | 14.82 | 27.22 |
| - | ✓ | - | - | 0.78(↑0.39) | 1.70(↑0.94) | 23.18(↑8.36) | 44.90(↑17.68) |
| ✓ | - | - | - | 0.41 | 0.80 | 15.52 | 28.50 |
| ✓ | ✓ | - | - | 1.09(↑0.68) | 2.33(↑1.53) | 32.42(↑16.9) | 65.43(↑36.93) |
| ✓ | ✓ | ✓ | - | 0.96(↓0.13) | 2.12(↓0.21) | 30.10(↓2.32) | 60.20(↓5.23) |
| ✓ | ✓ | ✓ | ✓ | 1.19(↑0.77) | 2.48(↑0.36) | 31.56(↑1.46) | 64.52(↑4.32) |

## 4.5 Ablation Study

We ablate some key components of our implementation for both ADE/FDE and APD/FPD metrics, see Table 4 and Table 5. (1)**Prediction clustering** is a common post-processing method, which improves the ADE/FDE as expected. However, it hurts the diversity for nomalizing flow model with single Gaussian prior. This is reasonable as the single Gaussian prior tends to generate trajectories densely close to the most likelihood and prediction clustering can't cluster them into well-separated clusters for different motion intentions. (2)**Mixed Gaussian prior** help the model generates more diverse outputs and achieves higher APD/FPD scores and this improvement can be further enhanced by prediction clustering. It also increases ADE/FDE scores a lot, we believe this is because mixed Gaussian prior relieves the difficulty of learning the Jacobians for distribution transformation. Thus more under-explored patterns, which may be selected as the "best-of-$M$" samples in rare but plausible scenarios, have the chance to be expressed. (3)**Learnable variance** improve ADE/FDE while bring down APD/FPD a bit. We find that the learnable variance usually converges to a smaller value than the fixed situation. This is encouraged by the supervision from the ground truth (most likelihood) to a desired steeper Gaussian, thus hurting the diversity. However, its substantial improvement in ADE/FDE indicates that it remains a valuable component of the model architecture. (4)**Inverse loss** provides a straightforward supervision of the trajectory in the coordinate space, which is also proved beneficial for ADE/FDE and APD/FPD scores.

## 5 Conclusion

We focus on improving the diversity while keeping the estimated probability tractable for trajectory forecasting in this work. We noticed the poor expressiveness of Gaussian distribution as the original sampling distribution for normalizing flow-based methods to generate complicated and clustered outcome patterns. We thus propose to construct a mixed Gaussian prior to help learn Jacobians for distribution transformation with less difficulty and higher flexibility. Based on this main innovation, we propose Mixed Gaussian Flow (MGF) model for the diverse and controllable trajectory generation. The cooperating strategy of constructing the prior distribution and training the model is also designed. According to the evaluation of popular benchmarks, we demonstrate that MGF achieves state-of-the-art prediction alignment and diversity. It also has other good properties such as controllability and being invertible for probability estimates.

## Acknowledgments

This project is funded in part by the Centre for Perceptual and Interactive Intelligence (CPII) Ltd under the Innovation and Technology Commission (ITC)'s InnoHK. Dahua Lin is a PI of CPII under the InnoHK. This project is also supported by Shanghai Artificial Intelligence Laboratory.

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

# A Inconsistent Practice of the Evaluation for Trajectory Prediction

Previous research in the area of trajectory forecasting, i.e. trajectory prediction, has focused on multiple datasets for quantitative evaluation. However, we notice that the evaluation settings of previous works are inconsistent thus making noisy and unfair comparisons in the benchmarks we usually refer to.

## A.1 Inconsistent evaluation convention on ETH/UCY

The ETH/UCY [24, 38] dataset, comprising five subsets of data, serves as the primary benchmark for human trajectory prediction. It is not proposed as a single dataset in the first place but merges many different resources. Therefore, there are no official guidelines for data splitting and model evaluation metrics. Consequently, previous studies employ different evaluation conventions and falsely confuse results from different conventions together for benchmarking.

The benchmark widely adopted by the research community was initially proposed by Social-GAN [13]. It adheres to the following principles:

1. Utilizing data with a sampling rate of 10 FPS in all subsets.

2. Employing a leave-one-out approach for splitting, where the model is trained on four subsets and tested on the remaining one subset.

3. Dividing the raw trajectory samples into specific train, eval, and test sets.

Subsequent works like Trajectron++ [46], AgentFormer [63], and EqMotion [59] have widely embraced this benchmarking and evaluation setting.

However, not all studies adhere to this benchmark. To identify examples, we conducted a review of recent open-access conference papers, which reveals the following divergence.

**1. Sampling Rate on ETH dataset.** There are two widely used sampling rates for the evaluation of the ETH dataset. While Social-GAN [13] utilized the data with a sampling rate of 10FPS (SR=10), other works, such as SR-LSTM [64], V2Net [55], SocialCircle [56], STAR [61], PCCSNet [51], Stimulus Verification [50], and MG-GAN [7], used the version with a sampling rate of 6FPS (SR=6). The SR=6 version contains more data, with a total of 8,908 frames, whereas the SR=10 version consists of only 5,492 frames. Based on our experience, the same model tends to yield higher evaluation scores (ADE/FDE) on the SR=6 version compared to the SR=10 version.

**2. Data Splitting.** Social-GAN follows a specific scheme and ratio for splitting the data into train/val/test sets, while some works have adopted different conventions. For instance, Sophie [45] selected fewer training scenes, MG-GAN [7] used the complete training scene data for training while separating a portion of the test set for evaluation. Also, works that choose the SR=6 version data in the ETH dataset adopt different splitting conventions, because they do not share the same raw trajectory data samples with Social-GAN, which uses the SR=10 version data.

**3. Inconsistent Data Pre-processing.** Some other studies, such as Y-Net [29], Introvert [48], and Next [27], provide processed data without the raw data and the processing scripts. The provided processed data for val/test sets does not align with the Social-GAN benchmark.

## A.2 Inconsistent evaluation convention on SDD

Compared to ETH/UCY, SDD [42] is a more recent dataset consisting of 20 scenes captured in bird's eye view. SDD contains various moving agents such as pedestrians, bicycles, and cars.

Most works follow the setting of TrajNet [44] which comes from a public challenge. However, some works adopt different evaluation way compared to TrajNet. SimAug [26] reprocesses the raw videos and gets a set of data files different from TrajNet's. Besides, it uses a different data splitting convention. Subsequent works such as V2Net [55] and SocialCircle [56] follow the same setting that SimAug starts. DAG-Net [34] shared the same data file with TrajNet, but used a different data splitting approach. Social-Implicit [33] followed its setting.

Therefore, there are multiple different evaluation protocol conventions on the ETH/UCY and SDD datasets. Because the data splitting for training/test and evaluation details are different, putting the evaluation numbers from them together provides misleading quantitative observations, which the community has been using for a while. We point it out and make a complete summary of these

misalignments, wishing future research aware of this to avoid potential continuity of the mis-practice and provide a fair comparison.

### A.3 Summary and Our Practice

When comparing baseline results, many previous studies fail to meticulously verify whether they adhere to the same convention, thereby leading to unfair comparisons. To compare the performance of various models fairly, we follow the practice that the community adheres to the most: Social-GAN's convention for ETH/UCY and TrajNet's convention for SDD.

More specifically, for ETH/UCY dataset, we recommend to use the preprocessed data and dataloader from SocialGAN [13]/Trajectron++ [46]/AgentFormer [63]. For the SDD dataset, we recommend using the preprocessed data and dataloader from Y-Net [29]. Although their data processing methods may differ, they share the same data source and data splitting approach, facilitating fair comparisons.

We also note that, in the early version of Trajectron++, a misuse of the *np.gradient* function during computation resulted in the model accessing future information. Rectifying this bug typically leads to a significant decrease in scores. Consequently, several Trajectron++-based studies have achieved improved scores.

## B Sensitivity of diversity metrics

As we all know, the ADE/FDE employ a "best-of-$M$" computation approach, where $M$ significantly influences the results: larger $M$ values yield lower ADE/FDE scores. We wonder whether APD/FPD metrics demonstrate similar sensitivity to $M$. Since APD/FPD calculations consider all trajectories collectively, intuitively, their values would remain stable with varing $M$. Our experimental results confirmed this, see Table 6. The APD/FPD metrics for FlowChain and MGF(w/o Pred. Clustering) remain stable as $M$ increases, whereas MGF exhibits a slight decrease. This decrease can be attributed to MGF's prediction clustering mechanism, which initially generates fixed $J$ samples (e.g., here J=500) and then clustering them into $M$ output trajectories. As $M$ increases, the impact of prediction clustering gradually diminishes (when $M = J$, prediction clustering is deprecated). Given that prediction clustering contributes to improving diversity in our ablation study, its weakening effect leads to a monotonic decrease in APD/FPD

Table 6: APD/FPD performance of FlowChain and MGF on ETH/UCY and SDD dataset when M varies.

| M | FlowChain | | MGF | | MGF(w/o Pred. Clustering) | |
|---|---|---|---|---|---|---|
| | **ETH/UCY** | **SDD** | **ETH/UCY** | **SDD** | **ETH/UCY** | **SDD** |
| 10 | 0.37/0.72 | 14.05/25.88 | 0.97/2.18 | 31.96/65.54 | 0.75/1.66 | 24.11/47.60 |
| 20 | 0.39/0.76 | 14.81/27.23 | 1.02/2.28 | 31.50/64.39 | 0.81/1.80 | 24.84/48.78 |
| 30 | 0.40/0.77 | 15.05/27.68 | 0.99/2.23 | 30.62/62.29 | 0.83/1.83 | 26.56/52.26 |
| 40 | 0.40/0.78 | 15.18/27.91 | 0.97/2.16 | 29.85/60.47 | 0.84/1.87 | 26.33/51.72 |
| 50 | 0.40/0.78 | 15.25/28.03 | 0.95/2.11 | 29.46/59.45 | 0.84/1.86 | 26.16/51.43 |
| 60 | 0.40/0.78 | 15.29/28.10 | 0.93/2.07 | 29.00/58.34 | 0.84/1.87 | 26.16/51.33 |
| 70 | 0.40/0.79 | 15.31/28.12 | 0.91/2.03 | 28.78/57.75 | 0.84/1.86 | 26.20/51.53 |
| 80 | 0.40/0.79 | 15.34/28.19 | 0.90/2.01 | 28.54/57.10 | 0.84/1.87 | 26.48/52.20 |

## C Controllable generation by data augmentation

Besides the fashion of manipulating generation results given the good property of our constructed mixed Gaussian prior in the main paper, we also use data augmentation to alter the data patterns in our training set, thereby obtaining different priors. This enables our model to fix corner cases that are difficult to handle with traditional flow-based models like FlowChain. Taking Figure 6 as an example, to generate the clusters representing green "U-turn", purple "left-turn", and cyan "right-turn" clusters in the right-middle figure, we duplicate and rotate the original future trajectory data by 180°, 90°, and -90°, respectively. Subsequently, we mix these rotated data with the original data in a fixed ratio to produce the augmented dataset (in this particular case, a 2:2:1:1 ratio is employed for the original:180°:90°:-90°). Then we apply k-means to the augmented dataset, thereby obtaining the new augmented prior distribution (depicted in the right-middle figure). Finally, we train the model using this augmented dataset. Compared to the generated results from FlowChain, after using the augmented data to construct the mixed Gaussian prior, our method can generate the

Table 7: minADE/minFDE of worst-$N$ predictions on UNIV dataset. By adding augmented data along with their corresponding cluster centers, our method significantly improves the performance on the corner cases.

| $N$ | FlowChain | | Augment-MGF | |
|---|---|---|---|---|
| | **ADE** | **FDE** | **ADE** | **FDE** |
| 10 | 3.13 | 6.54 | 0.75 | 1.30 |
| 50 | 2.40 | 4.90 | 0.90 | 1.57 |
| 100 | 2.06 | 4.29 | 1.07 | 1.86 |

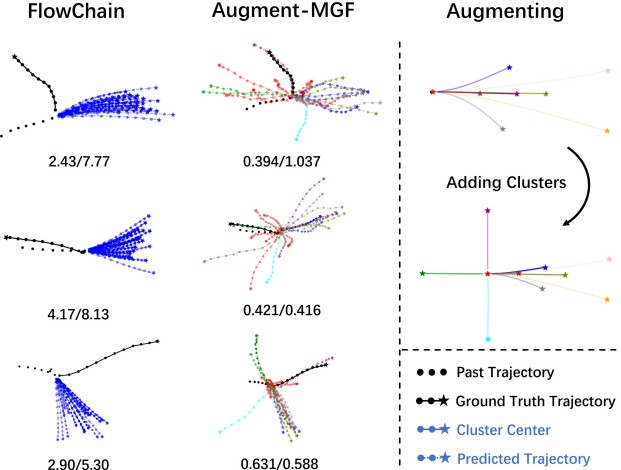

Figure 6: By adding augmented data along with their corresponding clusters into the construction of the mixed Gaussian prior, we could manipulate the generation patterns as we desire. For example, we could inject some under-represented trajectory patterns. Then the model can generate corner cases that existing models fail to generate in a reasonable probability. We selected three examples from the UNIV dataset, namely sharp left/right turns and U-turns. The left column shows the predictions from FlowChain, while the middle column shows the predictions of MGF with augmented priors

under-explored trajectory patterns with a higher chance. After this manipulation, we could change the mixed Gaussian prior as we desire, such as amplifying the chance of generating corner cases in this example.

On the other hand, we quantitatively evaluate the ability to generate under-represented trajectory patterns. Table 7 compares the ADE/FDE scores of their worst-$N$ samples on the UNIV dataset. Typically, the samples from the test set with the worst ADE/FDE relate to the under-represented corner cases of future trajectories. The results demonstrate quantitatively that MGF can better generate the under-represented motion patterns after injecting the desired corner cases on purpose by manipulating the mixed Gaussian prior as mentioned above. We note that all the provided examples of manipulating the mixed Gaussian prior to controlling the generation statistics do not require fine-tuning or any operation to the normalizing flow itself. As manipulating the mixed Gaussian prior is purely a parameter updating processing without any training and gradient backpropagation, all the manipulation is very fast in practice. This suggests the good efficiency and flexibility of our proposed method to achieve controllable generations.

## 6    Limitations

Limited by computing resources, we did not utilize the map information in our model. Some generated trajectories may overlap with obstacles, thus decreasing the upper bound of MGF's ability. Also, we found that agents can occasionally collide with each other due to the limited ability of the history encoder. Future works may take more consideration to the collision among agents or between agents and the environment.

