# OpenReview forum: "MGF: Mixed Gaussian Flow for Diverse Trajectory Prediction"
_NeurIPS.cc/2024/Conference — NeurIPS 2024 poster_

### Official Review · Reviewer_pmqe · 2024-06-21

**Soundness:** 3
**Presentation:** 2
**Contribution:** 1
**Rating:** 3
**Confidence:** 4

**Summary:**

This paper proposes to solve the trajectory prediction task by normalizing flow based model with mixture Gaussian assumption. The trajectories are clustered to multiple Gaussian distributions in the pre-processing stage over training data. During inference, priors are sampled from the mixture Gaussian distribution as the guidance of trajectory prediction. Moreover, new metrics are introduced to evaluate the diversity of multiple trajectory generation.

**Strengths:**

1. The authors analyze the limitations of current metrics ADE and FDE, which only focus on trajectory level prediction accuracy.

2. The idea of controllability for trajectory prediction based on learned mixture Gaussian distribution is interesting, although I do not think it is an appropriate task to demonstrate the controllability (see details in Questions)

**Weaknesses:**

1. The mixture Gaussian assumption is not accurate.

    Human trajectories are always affected by the interaction with other pedestrians and the environment. I agree that the possibility of trajectory conditional on i) historical trajectory ii) trajectory of other pedestrians iii) environment (e.g. maps) could be a mixture Gaussian distribution. However, all trajectories follow mixture Gaussian distribution without any conditions may not be an good assumption. Because the scenes in training and testing set are different. The learned distribution only fit on the training set with specific scenes, which cannot be generealized to different scenes.


2. The proposed diversity metrics is not reasonable.

    The diversity in multiple trajectory generation is necessary only on the premise that the generated trajectories make sense. The definition of APD and FPD only consider how different amont the generated trajectories but ignore whether the trajectories are reasonable or not. For example, 20 future trajectories are manually set to equally cover all directions from time step t_1, which will have very good score on APD and FPD but we cannot say these future trajectories are good predictions.


3. This paper ignores lots of recent works in SOTA comparison. E.g.
    Mangalam, Karttikeya, et al. "From goals, waypoints & paths to long term human trajectory forecasting." ICCV. 2021.

    Weibo Mao, et al.. Leapfrog diffusion model for stochastic trajectory prediction. CVPR, 2023.

    Jiangbei Yue, et al. Human trajectory prediction via neural social physics. ECCV, 2022

    Yonghao Dong,  et al. Sparse instance conditioned multimodal trajectory prediction. ICCV, 2023.

    ...

    Although authors mentioned that for some of the recent works, the processed data is slightly different. It would be more convincible to comapre to them with same training/testing data instead of ignoring these works as these works represent the most recent progress compared to original benchmarks such as socialGAN.

**Questions:**

Please see the questions in weaknesses.

Besides, here are some suggestions.

1. The readability of notations and equations can be improved. For example, the paper does not include the scene map modeling, then map M (line 111) is useless. If O_{t0:t1} (line 119) only contains map information, then O can also be excluded from equations.

2. This task may not be the appropriate task for the proposed method.

    Although the task of trajectory prediction contains multiple trajectory output which usually involves generative models, this task is not a pure generation problem. In this task, the accuracy is still with highest priority. For example, a self-driving vehicle need to predict a pedestrian's (or another vehicle's) future trajectory to avoid collision, where only the correct trajectory is required. Since motion intentions of agents cannot be determined by their historical positions (as described in introduction), multiple predictions are necessary to cover the most possible futures instead of multiple predictions as diverse as possible (or controllable predictions).

    I believe it would be more appropriate to evaluate the proposed method in the tasks such as motion simulation in synthetic data, where diversity and controllability motion trajectory is useful to make the agents look real.

**Limitations:**

Authors discussed the limitations in the appendix.

---

> ### Author Rebuttal · Authors · 2024-08-07
>
> **Mixed Gaussian assumption:** Our motivation is the difficulty of transforming a single-modal and symmetric original distribution to a complex target distribution by normalizing flow. We seek a solution to relieve the difficulty. We choose the problem setting to better provide a fair quantitative evaluation protocol. In the area of trajectory forecasting, the problem setups are different. Most works do not consider the map information (at least on the benchmarks of ETH/UCY and SDD) while some others consider the map information. We agree that map information and other factors can be important in practice, but we focus on the simple convention without considering them because (1) we follow the convention with the most baseline works to compare with; (2) our innovation is mainly about enhancing the expressiveness of complicated target distribution of normalizing flow. We would like to choose the setting with the minimum necessary factors to avoid irrelevant inductive bias, such as map information encoder design in different baseline works. Adding other conditional information will steer our focus to a different problem setting, though worth studying for practical purposes, which may confuse the contribution and intention of our work.
>
>
> **APD/FPD not reasonable:** Without other information, such as the obstacle in the scene, we can't judge if a trajectory is `reasonable` or not. Our focus in this work is to design a method with a better diversity of the generation results by normalizing flow. The fundamental is to enhance the expressiveness when transforming to a complicated target distribution. I believe that can't say that the long-standing task of motion generation/forecasting is meaningless if no scene information is considered as it is a good platform to study the generative models' capacity as a probabilistic time series prediction task. Further adding other information can make the solutions from this task setting better fit with practical situations.
>
> We agree that a single APD/FPD is not a perfect metric, it should be used along with the ADE/FDE. For example, ``equally cover all directions`` as a solution can reach high APD/FPD, but would probably cause low ADE/FDE which are the main metrics in the area of trajectory forecasting for many years.
>
> The APD/FPD metrics are more inspiring through the new ablation provided in the pdf file (Table 1&2). It highlights that mixed Gaussian prior can significantly increase the diversity compared to the single Gaussian prior, while also boosting the accuracy measured by ADE/FDE even though the two sets of metrics are usually considered a hard trade-off.
>
>
> **Missing recent works:** We appreciate the mentioned recent related works. As we emphasize in the appendix, it is chaotic for benchmarking in the area of trajectory forecasting that different methods use different data pre-processing and evaluation conventions but compare together. We made intensive efforts to present the different conventions and choose the most widely adopted one for our benchmarking. Please refer to Appendix A in the supplementary for details. We did not include some mentioned related works in the main paper due to this reason:
> 1. Ynet[1] and Yue, et al[2] use a different data convention from the widely adopted Social-Gan data convention. We explain this in Appendix A (Lines 575-577). So we can't include them in the benchmark.
> 2. [3] only provides NBA dataset processing details. In their GitHub code, they only provide the training and evaluation on the NBA dataset. It is not able to know the details of evaluation/training on the ETH/UCY dataset.
>
> There are some recent works that we should have included, such as [4,5]. We will add these works to the comparison.
>
> We agree that it would be more convincible to compare more related works following their separate data pre-processing and evaluation conventions. However, this requires us to train and evaluate our model in at least seven different evaluation protocols. We choose the benchmarking convention following Social-GAN which includes most previous arts as we believe it has the best representativeness. Moreover, many works such as [3] provide no details, code, or processed data for us to know their data processing and evaluation protocol. Therefore, this is impossible for us to compare with these works fairly. We may benchmark under other data conventions if enough baselines can be gathered to provide a more complete sense of our method's performance.
>
> > [1] "From goals, waypoints & paths to long term human trajectory forecasting"
>
> > [2] "Human trajectory prediction via neural social physics` share the same data with Ynet"
>
> > [3] "Leapfrog diffusion model for stochastic trajectory prediction"
>
> > [4] "Sparse instance conditioned multimodal trajectory prediction"
>
> > [5] "LG-Traj: LLM Guided Pedestrian Trajectory Prediction"
>
>
> **Not an appropriate task:** Our goal is to design a generative model with good controllability, diversity, and invertibility and it is capable of modeling a complicated distribution with imbalanced training data. We understand the concern that diversity is not a high priority for practical trajectory forecasting cases (again, this goes back to the different opinions about whether the problem formulation is proper or worth studying). But we believe that it is a proper task for us to demonstrate our method because (1) the `best-of-M` evaluation protocol in this area provides the focus on both accuracy and diversity therefore capable of estimating whether our method can generate diverse results with faithful probability; (2) the task of trajectory prediction is a long-standing task in the computer vision community, thus provides a rich line of baselines to compare with. However, we agree that some other tasks, such as motion simulation and animation in a virtual environment, can also be good. They would be interesting future works to migrate our method into the graphics and animation communities.

---

> ### Author Response · Authors · 2024-08-12
> **Follow-up meesage**
>
> Dear Reviewer pmqe,
>
> Please let us know if our responses have addressed your concerns or if you have any questions or concerns regarding our submission. We are glad to further discuss them. Thanks!

---

> ### Author Response · Authors · 2024-08-13
> **More discussion about the task setting**
>
> Dear Reviewer pmqe,
>
> We understand your concern that our task setting can't be fit to certain user scenarios directly, such as auto-driving trajectory planning. We would like to elaborate more about it to clarify potential misunderstandings.
>
> Our problem setting is not to directly solve the auto-driving problem and we didn't claim it. We follow a long-standing computer vision problem setting for trajectory prediction/forecasting. It has been widely recognized as a main task for studying probabilistic generation diversity. A line of works published on top-tier venues, such as [1,2,3,4,5,6,7,8] (there are many more unlisted), follow this task setting. They all study on the same dataset as we use in this submission (ETH/UCY or SDD). They are proposed to study the balance between (1) more aligned/accurate max-likelihood prediction and (2) more diverse probabilistic generation. Both goals are expected to be achieved from a single model in a balanced fashion. Focusing on a single goal only causes defective solutions, either overfit deterministic methods (if only caring about accuracy/alignment) or meaningless random generation (if only caring about diversity).
>
> The related works mentioned by Reviewer pmqe also emphasize that accurate and diverse trajectory generation is a main contribution of their method. For example:
> 1. [6] claims its main contribution as `achieves precise and diverse predictions with fast inference speed`.
> 2. [7] claims its contribution as ` exploits the proposed epistemic & aleatoric structure for diverse trajectory predictions across long prediction horizons`.
> 3. [8] claims its main contributions including `a balanced solution, i.e., a novel sparse instance, ...to guide the multimodal trajectory prediction...for the multimodal trajectory
> prediction`.
>
> If you have any concerns not addressed, please let use know. The authors will be pleased for follow-up discussions.
>
> Reference:
> > [1] "AgentFormer: Agent-Aware Transformers for Socio-Temporal Multi-Agent Forecasting", ICCV'2021
>
> > [2] "Remember Intentions: Retrospective-Memory-based Trajectory Prediction", CVPR'2022
>
> > [3] "Stochastic Trajectory Prediction via Motion Indeterminacy Diffusion", CVPR'2022
>
> > [4] "Progressive Pretext Task Learning for Human Trajectory Prediction", ECCV'2024
>
> > [5] "Can Language Beat Numerical Regression? Language-Based Multimodal Trajectory Prediction", CVPR'2024
>
> > [6] "Leapfrog Diffusion Model for Stochastic Trajectory Prediction", CVPR'2023
>
> > [7] "From goals, waypoints & paths to long term human trajectory forecasting", ICCV'2021
>
> > [8] "Sparse instance conditioned multimodal trajectory prediction", ICCV'2023

---

### Official Review · Reviewer_1BoH · 2024-07-05

**Soundness:** 3
**Presentation:** 2
**Contribution:** 3
**Rating:** 7
**Confidence:** 4

**Summary:**

Due to the asymmetric and multi-modal nature of future trajectories, the authors point out that the standard Gaussian prior with a neural network-based transformation is insufficient for probabilistic trajectory prediction. They propose Mixed Gaussian Flow (MGF), a method that uses the mixed Gaussian prior in the normalizing flow model. MGF allows diverse and controllable generation by leveraging the prior distribution. A minimum-of-M loss is designed for the inverse process to increase the predicting precision. Substantial experimental results on well-known datasets like ETH/UCY and SDD verify the effectiveness of MGF.

**Strengths:**

S1. The motivation is strong and the method is intuitive.

S2. MGF allows controllable generation by manipulating the prior distribution. It also secures under-represented future trajectory patterns.

S3. The experimental results are solid and convincing.

**Weaknesses:**

W1. This paper challenges the standard normal prior assumption in the normalizing flow model, but lacks in theoretical support. Also see Q6.

**Questions:**

Q1. The prior distribution is obtained by clustering trajectories in the training set. Will it suffer from a distribution shift in the testing set? Specifically, in ETH/UCY dataset, the testing set contains scenes that do not appear in the training set. Can MGF deal with this issue? If not, can you figure out any possible solution?

Q2. The asymptotic behavior of ADE/FDE is well-studied [Ref.1]. Is it possible to theoretically or empirically analyze the asymptotic behavior of APD/FPD for a sufficiently large $M$? Is the value of APD/FPD sensitive to $M$?

[Ref.1] Analyzing the Variety Loss in the Context of Probabilistic Trajectory Prediction. In ICCV, 2019.

Q3. Some other papers have proposed metrics for measuring the diversity of trajectories, like the average maximum eigenvalue [Ref.2] and the energy score [Ref.3]. Can you discuss the connection and difference of APD/FPD with these metrics? Is it possible to evaluate the trajectory diversity of different methods using these metrics?

[Ref.2] Social-implicit: Rethinking trajectory prediction evaluation and the effectiveness of implicit maximum likelihood estimation. In ECCV, 2022.

[Ref.3] Evaluation of Trajectory Distribution Predictions with Energy Score. In ICML, 2024.

Q4. In Section 4.4, how do you edit the directions of the cluster means?

Q5. Can you provide some empirical evidence for the claim of "limited and imbalanced data" in Line 30 of Section 1, e.g., the weights of each Gaussian component of the prior distribution?

Q6. Is it possible to find some theoretical support for the claim that "deriving a desired complex distribution from a simple and symmetric prior distribution is challenging, especially with limited and imbalanced data" in Line 29-30 of Section 1?

Q7. In Line 230-232, the authors say "Instead, it is hard to back-propagate the gradient through the sampling process of a Gaussian mixture with the reparametrization applied." I think the gumbel-softmax trick can be used for selecting components while allowing back-propagation. What do you mean by saying "it is hard to back-propagate"?

Comment.

C1. The authors clearly describe the subtle differences in evaluation in existing methods. It really saves my time and helps researchers in trajectory prediction.

C2. Appendix B contains some important facts. Please summarize these discoveries in the main text.

**Update:**

I have read the authors' response. They have addressed most of my concerns. I decide to raise my rating to 7. I hope that the authors can better organize the paper so that the presentation is clearer and it is easier for the readers to grasp the paper's significance.

---

> ### Author Rebuttal · Authors · 2024-08-07
>
> **Theoretical support behind our claim:** Great suggestion! Our claim about the difficulty of training normalizing flow to transferring a naive and symmetric distribution, e.g., simple Gaussian, to a complex distribution, is mainly from our empirical observations. But there are some insights behind this observation:
> 1. VAE or Diffusion learns a conditional distribution to sample outcomes, whose representation is low-dimensional. For example, only $\mu$ and $\sigma$ as the output of VAE. On the other hand, normalizing flow is trained to learn a Jacobian between distributions, requiring a high-dimensional generation capacity.
> 2. Given the same input noise, VAE or Diffusion will generate different outcomes in multiple rollouts, only required to be constrained in an underlying distribution. However, normalizing flow has to do a deterministic transformation from the input noise to a certain sample in the target distribution.
> 3. Normalizing flow is typically trained with a single ground truth. With the naive Gaussian prior, if the target distribution is complex and multi-modal, we need to transform the single mode in the original distribution to more than one target mode. This causes a contradiction between training gradients.
> 4. Normalizing flow is designed to be inversible. To learn a Jacobian between a single-modal original distribution and a complex/multi-modal target distribution, the one-to-many mapping also causes difficulty when training inversely.
>
> There are many discussions about the difficulty of training normalizing flow in practice to represent a complex target distribution, such as[1,2,3]. In [3], normalizing flow is considered inefficient and difficult to train due to invertibility constraints on transformations and as a consequence input and output spaces with the same dimension. This matches our discussion in the insight #1 above.
>
> We have to admit that normalizing flows is usually not as good as GAN or diffusion for high-dimensional data generation, such as images. However, its unique property of invertibility is irreplaceable for controllable and explicable generation as we desire in this work.
>
> > [1] "Representational aspects of depth and conditioning in normalizing flows"
>
> > [2] https://distill.pub/2019/gan-open-problems/#tradeoff
>
> > [3] "Deep Generative Modelling: A Comparative Review of VAEs, GANs, Normalizing Flows, Energy-Based and Autoregressive Models"
>
> **Distribution shift:** The problem of distribution shift exists in all supervised learning arts. We believe the common and shared intentions across data are important to adapt to test scenes. And the history encoder allows MGF to adjust generated trajectories based on the input historical trajectories. For example, if there is a group of people located in front of the agent, trajectories of “go straight” sub-Gaussian will be transferred into left/right/other trajectories to avoid collisions. As long as the shared intentions such as "move to avoid collision" hold, our work can generate reasonable trajectories in new situations.
>
> **The influence of the value of M to APD/FPD:** We add the ablation study about the value of $M$ in Table 3 in the submitted pdf file with $M$ increasing from 10 to 80. According to the results, APD/FPD is not sensitive to the value of $M$.
>
>
> **Comparison to other diversity metrics:** Here we compare the mentioned diversity metrics with ours:
> 1. AMD (Average Mahalanobis Distance) and AMV(Average Maximum Eigenvalue) are proposed in [4]. AMD computes the Mahalanobis distance to measure how close the generated samples are to the ground truth, similar to ADE. AMV quantifies the overall spread of the predictions for diversity measuring. It requires the confidence of predictions, which limits its application in many cases where no confidence or probability is available for predictions.
> 2. The metrics of ED and EI are proposed in [5]. ED measures accuracy and EI shares the same formulation with APD.  We note that [5] is accepted to ICML'2024 and was not publicly available yet at the due date of NeurIPS submission.
>
> > [4] "Social-implicit: Rethinking trajectory prediction evaluation and the effectiveness of implicit maximum likelihood estimation"
>
> > [5] "Evaluation of Trajectory Distribution Predictions with Energy Score."
>
> **Edit the direction of the cluster:** The cluster center is represented as a vector in shape $12 \times 2$, which is the mean value of the corresponding sub-Gaussian distribution, sharing the same dimensionality as the output trajectory. We can directly manipulate the value of this vector to represent different motion intentions.
>
>
> **Empirical evidence of the data imbalance:** Existing trajectory forecasting datasets contain imbalanced data where most trajectories are going straightforward. In Figure 4 in the paper, the weights of each cluster (represented by colors) are: Blue: 0.28, Brown: 0.09, Red: 0.12, Green: 0.1, Yellow: 0.16, Cyan:0.19, Purple: 0.05. This suggests very imbalanced data distributions for different motion intentions. Moreover, across all five subsets in the ETH/UCY datasets and one set in the SDD dataset, similar data patterns are observed: approximately 30% of the trajectories exhibit a 'stay' behavior, while 50-60% correspond to 'go straight' trajectories with varying speeds. Conversely, only 10-20% of the trajectories involve 'turn left/right' maneuvers, with 'U-turn' and other corner cases being extremely rare occurrences.
>
> **Difficulty of back-propagation:** After sampling a noise point **x** from a mixture of Gaussian **D**, it is non-trivial to back-propagate the gradient from applied to **x** to the learnable parameters of **D**.
> [6] has addressed this issue using approximations. It requires substantially more computation. Gumbel-Softmax trick deals with the problem of “sampling discrete data from a categorical distribution”, it is not a viable solution for our case.
>
> > [6]: https://github.com/vsimkus/torch-reparametrised-mixture-distribution

---

> ### Comment · Reviewer_1BoH · 2024-08-08
>
> Thanks for your response. The literature on normalizing flow is also helpful for me. You may consider adding descriptions of editing the cluster directions and the empirical evidence of data imbalance to the revised paper. For distribution shift, according to your description, I tend to think that clustering based on relative positions other than absolute positions will help learn transferable moving patterns. I would like to point out that:
>
> 1. From my understanding, AMD and AMV are solely based on sampled trajectories. Confidence outputs from probabilistic methods are generally not a prerequisite. EI and ED are recently proposed metrics. You do not have to compare your metrics with them according to the submission guidelines of NeurIPS. Still, I believe properly evaluating the performance of all methods on more metrics can help better understand their differences, which can be added to the revision or left for future work.
>
> 2. As I understand, selecting a component from a Gaussian mixture model is a process of sampling from a categorical distribution, where the gumbel-softmax trick can play a role in allowing back-propagation. I am not sure if it will be practical and effective and can be left for future work.
>
> Still, I have one minor concern.
>
> **The influence of the value of M to APD/FPD:** The results in Table 3 of the attached pdf show that APD/FPD is stable for FlowChain, yet a noticeable decrease can be observed for MGF on both ETH/UCY and SDD datasets as $M$ increases. Since APD/FDP are computed by taking the average over sample distances, it is expected that they will be stable for a sufficiently large $M$ (I may be wrong). This expectation and the results for FlowChain do not match the results for MGF even for $M=80$. Can you provide more explanation?

---

> > ### Author Response · Authors · 2024-08-10
> > **Reply to Reviewer 1BoH**
> >
> > **AMD & AMV:** Yes, by carefully check the [code implementation](https://github.com/abduallahmohamed/Social-Implicit/blob/main/amd_amv_kde_metrics.py) of AMD & AMV, we found the 2 metrics can be solely calculated based on sampled trajectories. The GMM can be fit by the internal computing process without confidence estimation. We will carefully review the recently proposed related metrics and add the them in the revised version of paper.
> >
> > **Gumble-softmax trick:** Really a good suggestion! Gumbel-softmax trick provides a solution[1,2,3] with approximation to derive a differentialble gradient when sampling from categorical distributions (in our case, the mixture of multiple sub-Gaussians). But applying the Gumble-softmax to learn a mixture of Gaussians in generative models has been reported difficult in practice in many cases[2,4] due to graident vanish. Still, adding Gumble-Softmax as a re-parameterization trick for graident BP would be an interesting component in the revised paper though that makes no impact to the main contribution of this work. We will leave it for future works.
> >
> > > [1] "Categorical Reparameterization with Gumbel-Softmax"
> > > [2] "Invertible Gaussian Reparameterization: Revisiting
> > the Gumbel-Softmax"
> > > [3] "Gradient Estimation with Stochastic Softmax Tricks"
> > > [4] https://gfchen01.cc/post/gmm_vae/
> >
> > **concern about APD/FPD:** Very good insight! Intuitively, the APD/FPD are expected to be stable for a sufficiently large M, as what FlowChain performs in the added ablation study (Table 3).
> > To add more detailed observations, the decrease of APD/FPD by MGF when M increases can be attributed to the use of prediction clustering in enhancing diversity.
> > In Table 3 in the general response pdf, predictions for M=10, 20, ..., 80 are obtained by first sampling 500 trajectories and then clustering them into M clusters. As M increases, the impact of prediction clustering gradually diminishes (when M=500, prediction clustering is deprecated). Given that prediction clustering contributes to improving diversity, its weakening effect leads to a monotonic decrease in APD/FPD until prediction clustering becomes entirely ineffective at M=500.
> > By removing the prediction clustering, we provide new results as shown in Table 4 below:
> >
> > *Table4: APD/FPD of MGF(w/o Prediction Clustering)*
> > | M   | ETH/UCY   | SDD         |
> > |-----|-----------|-------------|
> > | 10  | 0.75/1.66 | 24.11/47.60 |
> > | 20  | 0.81/1.80 | 24.84/48.78 |
> > | 30  | 0.83/1.83 | 26.56/52.26 |
> > | 40  | 0.84/1.87 | 26.33/51.72 |
> > | 50  | 0.84/1.86 | 26.16/51.43 |
> > | 60  | 0.84/1.87 | 26.16/51.33 |
> > | 70  | 0.84/1.86 | 26.20/51.53 |
> > | 80  | 0.84/1.87 | 26.48/52.20 |
> > | 100 | 0.85/1.88 | 26.51/52.07 |
> > | 200 | 0.85/1.89 | 26.74/52.49 |
> > | 300 | 0.85/1.89 | 26.65/52.36 |
> > | 400 | 0.85/1.89 | 26.72/52.47 |
> > | 500 | 0.85/1.89 | 26.72/52.50 |
> >
> > Without prediction clustering, APD/FPD of MGF remains stable as M increases, converging to approximately 0.85/1.89 on ETH/UCY and 26.7/52.5 on SDD.
> > Now, we observe that the when M increases, APD/FPD scores remain much more stable. This observation is well aligned with the intuition provided by you.

---

> > > ### Comment · Reviewer_1BoH · 2024-08-10
> > >
> > > Thanks for the recommending the blog and clarifying the details in metric evaluation. You can consider reporting both the results of Table 3 and Table 4 in th revised paper. I would consider rasing the rating to 7.

---

> > > > ### Author Response · Authors · 2024-08-11
> > > > **Reply to Reviewer 1BoH**
> > > >
> > > > We appreciate Reviewer 1BoH for raising the Gumble-softmax trick to us which can be an interesting add-on component to compare with in the paper. We thank Reviewer 1BoH for increasing the review score.
> > > >
> > > > We just got some preliminary experimental observations that we would need careful parameter study to avoid the gradient vanishing. We promise to conduct a discussion and empirical study about it in the revised paper.

---

### Official Review · Reviewer_qRNj · 2024-07-12

**Soundness:** 3
**Presentation:** 3
**Contribution:** 2
**Rating:** 6
**Confidence:** 4

**Summary:**

The authors proposed a new normalizing flow-based human trajectory prediction method called Mixed Gaussian Flow (MGF), which promotes diversity and controllability of prediction. The model uses a mixture of Gaussian model as the initial distribution to transform, rather than single-modal standard Gaussian distribution. The paper proposed using clustering of training data to mine modes of movements and use these modes to fit the initial sets of Gaussians for the mixture. The authors also claimed they proposed new metric measure diversity (APD, FPD). The authors have shown MGF can exceed SOTA performance on ETH-UCY and SDD datasets during their quantitative analysis, and showcase by controlling the mixture of Gaussian, the types of trajectory to generate can be controlled.

**Strengths:**

- The paper is well motivated, and by adopting Mixture of Gaussian, the model was able to achieve diversity and controllability
- The paper has shown normalizing flow-based HTP model can achieve SOTA performance with good design
- The model has good analysis (4.3, 4.4) showcasing the diversity and controllability of the model output.

**Weaknesses:**

- Line 112, 113 using $t_0, t_1, t_2$ might not be the best choice, the reviewer suggest maybe consider $t_0,t_1, \dots ,t_{c}, \dots, t_{T}$
- Suggest tidy up the variable representation of this section, for example using $X$ for observation sequence and $Y$ for future sequence
- Line 159 160, about normalizing flow overfitting, it would be great if the author could provide a reference to this claim or if this claim is by the authors themselves (also 129-130)
- Line 218-219, NLL loss is in sub-Gaussian, but how to decide the mixing coefficient (the reviewer is aware line 199 says it is decided from training data but specifically how?)
- Line 332, typo what is AED for APD or ADE?
- The ablation shows the ADE and FDE performance had a big increase with the prediction clustering, without prediction clustering MGF will have performance similar to AgentFormer and GroupNet and will not achieve SOTA, so does this means prediciton clustering/interntion clustering is decisive in accuracy performance of the model?
- The authors claimed they proposed APD metric, but it seems it is already in used in previous literatures under the same name and same formulation

@inproceedings{yuan2020dlow,
  title={Dlow: Diversifying latent flows for diverse human motion prediction},
  author={Yuan, Ye and Kitani, Kris},
  booktitle={Computer Vision--ECCV 2020: 16th European Conference, Glasgow, UK, August 23--28, 2020, Proceedings, Part IX 16},
  pages={346--364},
  year={2020},
  organization={Springer}
}

**Questions:**

- Abalation study was on ADE FDE only, but what is the effect on the diveristy? It would be great if the author could showcase this for the abalation, as diversity is the main contribution of the paper.
- If taking out mixture of gaussian and only use predictive clustering would the model achieve same level of ADE/FDE performance?
- Besides, this quesiton it would be great if the author could address the concerns in weakness section.
- The reviewer will consider increase the score if the authors could address and respond to the concerns mentioned above.

**Limitations:**

The reviewer appreciate the author mention in the supplementary about the limitation, where the model does not utilzed map information and predicitons will contain collision among agent. Although per guideline, the reviewer shall not punish this limitation, however, it is worth pointing out diversity without context might not be a sound logic. As solely promoting diversity might making the prediction output actually unrealistic considering the environmental and social context of the prediciton output. Although the author mentioned potential way to incorporating map information in the main paper, such proposal is more on the implicit side (as latent embeddings), which might not be able to address environmental collision issue.

---

> ### Author Rebuttal · Authors · 2024-08-07
>
> **Writing issues:** We appreciate the detailed suggestions about writing, we could consider them and improve the writing details. L332 should be ADE and FDE.
>
> **Normalizing flow overfitting:** We observe that normalizing flow tends to overfit to the single mode value (ground truth annotation) when using the naive Gaussian prior. This is because of the internal conflict between a probabilistic model and a deterministic and single ground truth supervision under the unimodal simple Gaussian prior. On the other hand, normalizing flow directly transfers a noise to a single outcome point, further causing the model to collapse to the max-likelihood value. As a comparison, VAE predicts a distribution where we can sample outcomes instead of a single outcome, relieving the overfitting issue with a reasonable value of distribution variance. Related discussion can be found in many previous literature, such as [1,2].
>
> > [1] "Noise Regularization for Conditional Density Estimation"
>
> > [2] "Why Normalizing Flows Fail to Detect Out-of-Distribution Data"
>
>
> **mixing coefficient:** In our implementation, we decide the coefficients by steps. We use k-means clustering to partition the trajectory data into clusters. The mixing coefficient of each sub-Gaussian corresponds to the weight of the corresponding cluster.
>
>
> **Prediction clustering to improve accuracy:** We add ablation experiments in Table 2 in the pdf file. The prediction clustering improves ADE/FDE with both single Gaussian and mixed Gaussian prior. Adding mixed Gaussian prior further boosts the performance.
>
> Moreover, we provide the results of ground truth alignment (ADE/FDE) to demonstrate that our method can output a single "likely" trajectory in a batch at a similar level as the best arts. Its flexibility and advantage of enhancing generation diversity is our main goal.
>
> **APD in Dlow:** Yes, we agree that a similar APD has been defined by Dlow[3]. We missed it as it was proposed on a different task and benchmarks. We will revise this paper and acknowledge where the metric was first proposed.
>
> > [3] "DLow: Diversifying Latent Flows for Diverse Human Motion Prediction"
>
> **Ablation studies on diversity:** Great suggestion! We add the ablation study to measure the impact of modules on diversity in Table 1 in the pdf file. There are two interesting observations:
> 1. The prediction clustering can significantly improve the diversity only with mixed Gaussian prior. With the naive Gaussian prior, it makes no contribution to the diversity. This is reasonable as the single Gaussian prior tends to generate trajectories densely close to the most likelihood and prediction clustering can't cluster them into well-separated clusters for different motion intentions.
> 2. The learnable var. slightly hurts the diversity. This is because the learnable var. usually converges to a smaller value than the fixed situation ($\sigma=1$). For instance, the average value is ~0.4-0.6 on ETH/UCY. This is encouraged by the supervision from the ground truth (most likelihood) to a desired steeper Gaussian, thus hurting the diversity. It is the trade-off between (1) aligning with the single likelihood provided by ground truth and (2) generating diverse outcomes that hide behind this observation.
>
> **Adding map information:** We agree that adding map information can be potentially more useful in realistic use cases. However it would lead to a different convention of benchmarking and the different processing of map information would introduce significant noise when comparing methods, especially considering many baselines have no open-sourced code or checkpoints. To focus on the main assumption and innovation in this work (a mixed Gaussian prior can improve the diversity of the normalizing flow-based trajectory forecasting), we decided not to use map information and follow the most widely adopted problem set up in this area. Adding map information would be an interesting future work but significant efforts are required to fairly compare different arts.

---

> ### Comment · Reviewer_qRNj · 2024-08-10
>
> The reviewer thanks the author for the response and the follow-up ablation studies. The author is able to address some of the original concerns and offers more insights on some of the questions.
>
> **Writing Issue**: The reviewer thanks the author for considering updating the manuscript and clarifying the typo.
>
> **Normalizing Flow Overfitting**: The reviewer thanks the authors' explanation and provide reference.
>
> **Mixing coefficient**: Thank you for the clarification, however, could you elaborate a bit more on the weight and how is it defined in the k-mean clustering? Is it based the number instance belongs to that cluster? Or other definition? Because general k-mean clustering does not usually involve weights.
>
> **APD originality issue**: The reviewer appreciates the author acknowledging this metric has been published before, however, these two tasks, although different in nature, are quite similar technically in their settings of the output. Please do make sure to revise the paper and give credit to the original literature, despite original work was on human motion prediction, motion trajectory and human trajectory could be both evaluated by APD metric, as the authors also chose to do so here, therefore it is not reasonable for the author to claim that metric as novel contribution here without mentioning the original work that it was first proposed.
>
> **Prediction clustering + Ablation:**
> - Indeed, using mixture of gaussian boosts ADE/FDE, compared to using stardard gaussian. Yet, the boost to ADE/FDE value with the addition of prediction clustering is significantly larger, the reviewer acknowledge ADE/FDE only reflect the best-of-m mode of the prediction, and does not reflect diversity or other quality of all model output. However, compare with switching to mixture gaussian as prior, ADE/FDE improvement by adding intention/prediction clustering is much more significant.
> - The reviewer really appreciates the author providing new abalation study on the effect of model component on diversity.  Indeed, the prediction clustering offers significant boost to APD/FPD to the model with mixture of gaussion prior (row 4 and row 5 in table 1) and less or no boost to even negative effect on APD/FPD (row1/row6 or row2/row7). However, if comparing row2 and row3, where the different is only between standard gaussian and mixture of gaussian as prior, the improvement on APD/FPD is less significant, especially on SDD dataset. Furthermore, in the main paper, the authors only offer APD/FPD benchmark against other model on ETH-UCY not SDD.
> - The learnable var. indeed slightly hurts diversity, however the difference is less significant compare with the point above and the reviewer thanks the author for the analysis.
> - Overall, although there is sign the mixture of gaussian as prior shows certain improvement of accuracy and diversity under specific circumstances, such improvement is less significant compared with the improvement brought by the prediction clustering. As a modeling choice, prediction clustering as a post-processing step is reasonable, however, since the author’s claim on mixture of gaussian improving sampling diversity is one of the main foci of the paper, more experiments to support the diversity claim on mixture of Gaussian are needed. One of such experiments would be to benchmark against other SOTA model on the SDD dataset, since the ablation on SDD especially shows mix signal and certain SOTA model such as AgentFormer is quite close in the diversity metric on the provided ETH-UCY diversity benchmark. The other suggested experiment would be showing more qualitative analysis to showcase through visualization of prediction outputs the effect of mixture Gaussian prior and prediction clustering. The reviewer understand such addition would be a significant amount of work, and might not be able to be added to the current submission.

---

> > ### Author Response · Authors · 2024-08-12
> > **Follow-up reply to Reviewer qRNj [1/2]**
> >
> > We appreciate the detailed feedback about our comments from Reviewer qRNj. To address Reviewer qRNj's concerns, we provide follow-up replies here.
> >
> > **Mixing coefficient**: The weight of a cluster is defined as the (number of trajectories belonging to that cluster)/(number of all trajectories).
> >
> > **APD originality issue:** Yes, though the two tasks are different, they are related. We promise that we will carefully review the related metrics and give full credit to the earlier proposal of the metric and the naming of APD.
> >
> > **Significance of prediction clustering and mixed Gaussian prior**:
> > First of all, we claimed the main contribution of this work as the proposal of mixed Gaussian prior to normalizing flow to enhance the generation diversity and not to hurt or even boost the alignment/accuracy.
> >
> > Second, to enhance the generation diversity, prediction clustering itself makes no significant contribution as observed in the ablation study. The mixed Gaussian prior is necessary for our method to achieve outstanding generation diversity (adding extra prediction clustering further boosts the diversity). Therefore, we are afraid that the reviewer's argument `"although there is sign the mixture of Gaussian as prior show certain improvement of accuracy and diversity under certain circumstances, such improvement is less significant comparing with the improvement brought by the prediction clustering."` is not accurate.
> >
> > In the previous ablations, the inverse loss is used together with mixed Gaussian priors and prediction clustering which can make potential noise to the comparison. We originally put prediction clustering as an optional component in the ablation study to provide more transparency though it is actually used in many existing works as a default module for post-processing.
> >
> > Now, we provide more ablation settings as in Table 5 and Table 6 below. We could have conclusions:
> > 1. Prediction clustering can't independently boost the generation diversity significantly.
> > 2. Using the Mixed Gaussian prior can improve both the diversity and the accuracy. Its improvement of diversity is much more significant than prediction clustering. Its improvement of accuracy/alignment is comparable to prediction clustering.
> > 3. Mixed Gaussian prior's improvements to diversity and accuracy/alignment can be stacked with prediction clustering.
> > 4. Learnable variance improves the accuracy but slightly hurts the diversity as we discussed in previous replies.
> >
> > *Table 5: More ablations about the generation diversity (APD/FPD, the higher the better)*
> > | Inv. Loss | Mixed Gauss. | Learnable Var. | Pred. Clustering | ETH/UCY   | SDD         |
> > |----------:|------------:|---------------:|-----------------:|-----------|-------------|
> > |           |             |                |                  | 0.39/0.76 | 14.82/27.22 |
> > |           |           √ |                |                  | 0.78/1.70 | 23.18/44.90 |
> > |           |             |                |                √ | 0.41/0.80 | 15.52/28.50 |
> > |           |           √ |                |                √ | 1.09/2.33 | 32.42/65.43 |
> > |           |           √ |              √ |                √ | 0.96/2.12 |  30.1/60.20 |
> >
> > *Table 6: More ablations about the prediction accuracy/alignment (ADE/FDE, the lower the better)*
> > | Inv. Loss | Mixed Gauss. | Learnable Var. | Pred. Clustering | ETH/UCY   | SDD         |
> > |----------:|------------:|---------------:|-----------------:|-----------|-------------|
> > |           |             |                |                  | 0.33/0.61 |  11.9/21.33 |
> > |           |           √ |                |                  | 0.29/0.48 |  11.38/19.28 |
> > |           |             |                |                √ | 0.29/0.54 | 10.63/18.80 |
> > |           |           √ |                |                √ | 0.27/0.48 |  9.19/15.86 |
> > |           |           √ |              √ |                √ | 0.23/0.39 |  8.71/14.86 |
> >
> > *Note: here we retrained the model to remove the inverse loss and make sure the same randomness for all entries in the tables.*
> >
> > The research of many computer vision tasks is always advanced by progressive improvement by continuous innovations. We believe as long as a method is effective in boosting performance and can be stacked with previous arts, it makes its own value to advance this area. Moreover, considering that our focus is to enhance the diversity of generation and prediction clustering can not actually boost the diversity, we believe our proposed mixed Gaussian priors have proven their unique value per the ablation studies.
> >
> > We note that inverse loss and prediction clustering are both techniques widely used in existing works in the area. We did not claim them as our contribution. To conclude, our innovation of Mixed Gaussian prior has been proven effective in improving both the diversity and the accuracy/alignment of the generation. The improvement can be stacked to the effect of existing tricks of inverse loss and prediction clustering.

---

> ### Comment · Reviewer_qRNj · 2024-08-10
>
> The reviewer acknowledges the contribution of mixture of Gaussians to boost the controllability of trajectory generation; its effectiveness of mining out different motion patterns out of the training data. The paper is well written and many intuitions and formulations are clearly explained. However, considering improving diversity as one of the main contribution of the paper, the experiments are showing mix signals and the diversity metric is only evaluated on one dataset for extensive benchmark against other SOTA models. The diversity metric is also not as original as the authors claimed. Plus, current submission lacks qualitative analysis showcasing the behavior of the different model component and model behavior under different type of inputs (for example, observation with slow speed vs. fast speed). Furthermore, the improvement of diversity alone might not be meaningful without considering the environmental and social factors in the prediction model, as social interaction and environmental constraint also dictates the behavior of human movement. Solely promote diversity without considering these constraints might not be a sound argument. Hence, the reviewer will keep the current score in terms of assessment.

---

> ### Author Response · Authors · 2024-08-12
> **Follow-up reply to Reviewer qRNj [2/2]**
>
> **Experiments on SDD:** In the original paper, we raised SDD in the paper to encourage further more intensive research on the challenging dataset. However, we had to put the dataset as secondary because of some practical difficulties:
> 1. SDD is a more challenging dataset than ETH/UCY with more complicated motion patterns and intentions. Also, SDD is a more recent benchmark. Many existing methods do not study SDD and just report results on ETH/UCY.
> 2. Among the baselines we included in the benchmark on SDD, only PECNet[1] provides the checkpoint to reproduce their reported results.
> 3. Otherwise, only MID[2] and MemoNet[3] open-source their training code (no pre-trained checkpoint is provided). Moreover, we unfortunately could not reproduce MID's result as reported in its paper by the released code. We could reproduce MemoNet's results on SDD close to its reported result.
>
> We encourage open-sourced research and transparent experiment settings on all datasets in this area. We would expect a more established benchmark on SDD with baseline methods. We wish to provide the diversity comparison for methods on SDD in the future version.
>
> > [1] "It is Not the Journey but the Destination: Endpoint Conditioned Trajectory Prediction"
>
> > [2] "Stochastic Trajectory Prediction via Motion Indeterminacy Diffusion"
>
> > [3] "Remember Intentions: Retrospective-Memory-based Trajectory Prediction"
>
> **Problem setups:** The task of trajectory prediction/forecasting has been studied for a long time in the community of computer vision. We followed the common problem setting of this long-standing task. According to the additional information on map/surroundings and social interaction:
> 1. Most works in this area do not take the map information into consideration. Moreover, adding extra map/environment information makes the task a totally different task, asking for a different line of benchmarking. Under such problem setups, more noise is included, especially the map information encoder, which is designed in different fashions in related works. This will confuse and blur our main focus in this work when seeking transparent and fair experimental evidence.
> 2. In our design, our method can actually capture the social relations of different pedestrians (by feeding the historical trajectory of all agents into History Encoder). Also, we did compare the methods modeling social interaction explicitly. Our method's performance is much better than many works that model social interaction explicitly, such as Social-GAN.
>
> **more qualitative results:** We did provide the visualization of prediction results in Figure 4 (in the paper) and Figure 6 (in the supplementary). Figure 6 is a good example to help understand the behavior and diversity difference between naive Gaussian prior and mixed Gaussian prior. We also provide illustration videos in the supplementary to understand the behaviors of different sub-Gaussians in our built model. We understand that Reviewer qRNj desires more quantitative visualization to assist in understanding the ablation study results. However, the qualitative difference is not significant for most pairs of comparison in the ablation study per our check. Moreover, we can't provide images or videos here per the rebuttal policy. We will consider the suggestion for the next version. Thanks.

---

> > ### Comment · Reviewer_qRNj · 2024-08-12
> >
> > We thank the author for the detailed response. the two new tables summarizing the ablation, especially the comparisons between first two rows are good demonstration of the  effectiveness of Mixture of Gaussian on the diversity aspect. Please add them to the final paper along with the analysis you provided in the second rebuttal. the reviewer believe that this concern is largely addressed.
> >
> > The reviewer still believe it is important to consider environment and social constraint, especially when discussing diversity, (as most of the works benchmarked in the main paper consider one or both factors), however, the proposal for mixture of Gaussian prior and the demonstration of effectiveness on controllability and diversity of trajectory prediction is certainly an important contribution and the reviewer recognize the merit of the paper. However, how to combine social and environmental factor into the current method is still an important future research direction. Therefore, the reviewer will raise the score and would strongly recommend the author discuss such limitation of the current work in the final paper. Please also make sure to address the APD issue and consider other previous suggestions.

---

> > > ### Author Response · Authors · 2024-08-13
> > >
> > > We appreciate Reviewer qRNj for the reply and raising the review score. We thank Reviewer qRNj for recognizing that the added ablation studies have addressed the concern. We will add them to the revised version of the paper.
> > >
> > > We agree that environmental information is important for trajectory forecasting. We will discuss it in the revised paper.
> > >
> > > Our proposed method does model the social constraint between agents though we didn't put the `social interaction` or similar terms explicitly in the paper. We would like to elaborate more about the social interaction/constraint referred to in the related works in this area for clarification. As stated in the `Implementation Details` in the paper, we follow the historical encoder as in Trajectron++[1]. Encoding and fusing historical trajectories of all or multiple agents is a typical way to model social interaction in this area. Some related works[1,2,3] explicitly claim social interaction modeling by using the same multi-agent historical trajectory encoder as ours. Moreover, there are some other genres of historical encoders used in this area to capture social interaction, such as LSTM encoder[4,5,6] and LSTM+GAT encoder[7]. Limited by the page limit and considering that social interaction is not a focus and contribution in this work (though MGF indeed models it), we didn't discuss it in the original paper. We will add a corresponding discussion about it in the revised paper and explicitly mention that our model handles social constraints between agents.
> > >
> > > We will definitely acknowledge the previously proposed APD metric and improve the paper according to previous suggestions in the paper revision.
> > >
> > >
> > > > [1] "Trajectron++: Dynamically-Feasible Trajectory Forecasting With Heterogeneous Data"
> > >
> > > > [2] "Fast Inference and Update of Probabilistic Density Estimation on Trajectory Prediction"
> > >
> > > > [3] "Stochastic Trajectory Prediction via Motion Indeterminacy Diffusion"
> > >
> > > > [4] "Social GAN: Socially Acceptable Trajectories with Generative Adversarial Networks"
> > >
> > > > [5] "Remember Intentions: Retrospective-Memory-based Trajectory Prediction"
> > >
> > > > [6] "Collaborative Motion Predication via Neural Motion Message Passing"
> > >
> > > > [7] "STGAT: Modeling Spatial-Temporal Interactions for Human Trajectory Prediction"

---

### Author Rebuttal · Authors · 2024-08-07

# General Response (GR)
We thank all the reviewers for their valuable suggestions and comments. We add new experiments in the separate pdf file to assist our responses to the reviewers' questions. We would also resolve the writing issues mentioned by reviewers in the paper revision.

---

### Decision · Program_Chairs · 2024-09-25

**Decision:**

Accept (poster)

**Comment:**

This paper introduces a normalizing flow (NF)-based approach for diverse trajectory prediction. In contrast to standard Gaussian priors used in NFs, the authors propose using a data-driven Gaussian mixture model (GMM), built from the analysis of trajectory patterns in the training dataset. The resulting Mixed Gaussian Flow (MGF) model is trained with both forward and inverse processes. The authors claim to include new diversity metrics, and experiments were conducted on the ETH/UCY and SDD datasets.

The paper initially received mixed reviews, with one weak acceptance recommendation (6), one weak rejection recommendation (4), and one rejection recommendation (3). The main concerns raised by the reviewers related to the justification of the method, the diversity metrics used in the submission, and several concerns regarding the experiments (e.g., relevance of the task, results with HD maps, comparison to other state-of-the-art methods). There was extensive discussion during the rebuttal period. The authors successfully addressed both theoretical and experimental concerns from reviewer 1BoH and reviewer qRNj, who respectively raised their ratings to 7 and 6, both recommending acceptance. However, reviewer pmqe maintained their initial recommendation, but did not respond to the authors' feedback.

The AC has thoroughly reviewed the submission and the discussions. The AC considers the problem of probabilistic forecasting to be highly important and appreciates the authors' efforts to fairly evaluate different methods. The proposed GMM data-driven prior is a simple yet relevant idea in the context of trajectory forecasting, and the forward and inverse processes are sound. In my opinion, a clearer positioning with respect to determinantal point processes (DPPs) [1] for representing diversity would have been beneficial. Additionally, the AC notes that the proposed diversity metrics have previously been introduced in the literature and urges the authors to properly credit related work—other diversity metrics, such as rF [2], have also been used. Therefore, the AC recommends acceptance but strongly encourages the authors to include the elements discussed during the rebuttal in the final paper.

[1] Diverse Trajectory Forecasting with Determinantal Point Processes. Ye Yuan, Kris Kitani. ICLR 2020.\
[2] Diverse and Admissible Trajectory Forecasting through Multimodal Context Understanding. S. H. Park, G. Lee, J. Seo, M. Bhat, M. Kang, J. Francis, A. Jadhav, P. P. Liang, and L. P. Morency. ECCV 2020.